# Intersecting social and environmental determinants of multidrug-resistant urinary tract infections in East Africa beyond antibiotic use

Katherine Keenan [1] ✉, Michail Papathomas[1], Stephen E. Mshana[2], Benon Asiimwe [3], John Kiiru[4], Andy G. Lynch [1], Mike Kesby[1], Stella Neema[3], Joseph R. Mwanga[2], Martha F. Mushi[2], Wei Jing[1], Dominique L. Green [1], Emmanuel Olamijuwon[1], Qing Zhang [1], Rachel Sippy [1], Kathryn J. Fredricks[1], Stephen H. Gillespie [1], Wilber Sabiiti[1], Joel Bazira[5], Derek J. Sloan[1], Blandina T. Mmbaga [6], Gibson Kibiki[7], David Aanensen[8], John Stelling [9], V. Anne Smith [1], Alison Sandeman [1], Matthew T. G. Holden [1] & HATUA Consortium*

The global health crisis of antibacterial resistance (ABR) poses a particular threat in low-resource settings like East Africa. Interventions for ABR typically target antibiotic use, overlooking the wider set of factors which drive vulnerability and behaviours. In this cross-sectional study, we investigated the joint contribution of behavioural, environmental, socioeconomic, and demographic factors associated with higher risk of multi-drug resistant urinary tract infections (MDR UTIs) in Kenya, Tanzania, and Uganda. We sampled outpatients with UTI symptoms in healthcare facilities and linked their microbiology data with patient, household and community level data. Using bivariate statistics and Bayesian profile regression on a sample of 1610 individuals, we show that individuals with higher risk of MDR UTIs were more likely to have compound and interrelated social and environmental disadvantages: they were on average older, with lower education, had more chronic illness, lived in resource-deprived households, more likely to have contact with animals, and human or animal waste. This suggests that interventions to tackle ABR need to take account of intersectional socio-environmental disadvantage as a priority.

Antibacterial resistance (ABR), continues to rise and is projected to be a leading cause of morbidity and mortality over the coming decades[1,2]. The burden of ABR is estimated to be higher in resource-limited settings like sub-Saharan Africa[2]. Of particular concern are multi-drug resistant (MDR) pathogens[3], which have high prevalence in low-and middle-income contexts (LMICs)[4,5] and are particularly challenging to treat. Most ABR interventions target behaviours to optimise antibiotic use[6], but the drivers of ABR stretch beyond biological or behavioural antibiotic use to encompass cultural, socioeconomic, political, infrastructural, and environmental components. These diverse drivers are

A full list of affiliations appears at the end of the paper. *A list of authors and their affiliations appears at the end of the paper.
✉e-mail: Katherine.keenan@st-andrews.ac.uk

often conceptualised as a system, a One Health problem, or an 'assemblage'[7–10]. Elements could also combine synergistically, or co-occur to produce intersectional vulnerabilities[11]. However very few studies investigate the joint contribution of these ABR drivers. One reason is the paucity of appropriately linked data measuring these factors. Another challenge is use of statistical approaches which try to mutually adjust factors, rather than capitalise on the interrelationships between them.

In this study, we address these gaps using a study design and methods which capture how diverse ABR drivers operate synergistically. Within a One Health framework we collected data across nine sites in East Africa to create a cross-sectional dataset which links environmental, social, economic behavioural and microbiological data. The study builds on evidence-based assemblage framework hypothesising how the drivers of UTI and ABR are interrelated[12]. We focus on multi-drug resistant urinary tract infection (MDR UTI), a growing problem in the region[5,13,14], and a condition which poses great clinical and public health threat. Our analysis addresses the following questions:

- How do environmental, social, economic, behavioural, and demographic factors relate to the burden of MDR UTI in East Africa?
- How are those factors jointly associated with a higher risk of MDR UTI?

## Existing research on the drivers of ABR and MDR

Antibiotic use in both humans and animals is considered a core biological driver of ABR and MDR, and a key target of interventions[15,16]. At individual level, use of antibiotics is associated with development of resistant infections[17]. In East African community settings, ease of access to antibiotics (ABs) in drug shops and pharmacies is understood to facilitate AB overuse and misuse and drive ABR[18–20]. Empiric prescription, driven by limited diagnostic capacity also drives indiscriminate use[21]. It is recognised that AB use or misuse is part of a larger system of vulnerability and risk[22] being correlated with other risk factors such as poorer health status, chronic disease, longer hospital stays, poor sanitation and close proximity to waste and animal products[23,24]. However, the contribution of these factors alongside AB use is rarely investigated in a holistic way.

Beyond antibiotic use, studies have revealed a range of demographic, clinical, behavioural, and socioeconomic factors that are associated with differential risk of ABR and MDR UTIs. Some of these are factors which confer vulnerability to UTI, including older age, gender, and poor health status[8,25,26]. Clinical history, such as recent surgery, wounds, long hospital stays, and medically invasive procedures are also associated with higher MDR UTI risk[25,27,28]. In low and middle income countries (LMICs) particularly, lower socioeconomic status[29] and structural dimensions of poverty[23,30,31] at various scales are suggested as broader drivers of ABR colonisation and infection. UTIs are commonly caused by gastrointestinal bacteria, therefore poor water, sanitation and hygiene (WASH) may increase both risks of infection and of spreading resistant gut-related bacteria[32]. Environmental One Health risk factors for ABR/MDR transmission include close proximity to livestock or aquaculture, or consumption of animal products, especially if those animals have been reared using ABs[8,18,29].

Current studies of these diverse drivers and their interrelations are limited. Ecological studies have modelled aggregate national rates of ABR, using regression-based approaches. These have identified important macro-level drivers such as poor WASH, political corruption, socioeconomic inequality, higher population density, and weaker regulatory standards[9,33]. Other studies have linked individual data across the human, environmental, and clinical domains. In resource-limited settings with many potential One Health drivers, this is challenging, so often studies are small-scale. For example, a household-based study in Tanzania, which investigated human carriage of resistant *Escherichia coli*, showed that cultural-ecological factors affecting transmission, including consumption and handling of milk and meat, were stronger drivers than recent AB use[34]. A longitudinal household study in Malawi found that prominent risk factors for ABR were advanced age, animals interacting with food, urban living, and the wet season[35]. These previous studies predominantly rely on regression methods that obscure complex interrelationships between variables. In this study, we address these gaps by employing standardised data across several East African sites and interrogating these using statistical methods capturing inter-relationships between risk factors and the outcome.

## Results

### Sample characteristics

Across the sample, the overall proportion of MDR UTI was 48%. This level this was proportionally lower in Kenyan sites and Nakapiripirit in Northeastern Uganda than other places (Fig. 1). Sites with the largest number of samples are in Kenya (Nairobi) and Tanzania (Mwanza). The univariate distribution of the variables and outcomes across the three countries is shown in Supplementary Data 1. Approximately half of the samples (51%) were collected from primary healthcare facilities, but this was more common in Ugandan and Tanzania samples than in Kenya. Three quarters of the patients lived in urban areas, and this was higher in both Kenya and Tanzania than Uganda. International travel was also more common in Kenyan and Tanzanian patient groups. Patients were most often aged 25–44 years, female, married, and working. Proportionally more Kenyan patients had secondary education than in Tanzania or Uganda (85% vs 23% and 28%). Treatment seeking behaviours for UTI (i.e., number of steps in seeking treatment, self-treatment, AB use, clinic attendance) was generally more complex in Tanzania and Ugandan patients than those in Kenya. Overall, 10% of the sample self-reported having a non-communicable disease (NCD), 5% reported having HIV/AIDS, and 56% had taken ABs in the previous 6 months. Household characteristics varied by country, but overall, most of the sample had protected sources of washing and drinking water, one third had livestock and around half had a form of health insurance. Participating households in Uganda and Tanzania were more likely to keep livestock, have shared toilet facilities or pit latrines, shared water facilities, and lower levels of asset ownership than those in Kenya.

### Bivariate associations with MDR UTI

The outcome of MDR UTI was significantly associated with 23 of the 67 contextual, household, and individual level variables (using an adjusted FDR $p$-value of < 0.05, Supplementary Data 2 and Fig. 2). MDR rates were higher among patients recruited in secondary and tertiary versus primary care (52% vs 43%). The proportion of MDR UTI was also higher among patients living in households that used manure for building, fertiliser, or fuel, that did not consume milk regularly, or were close to sites where rubbish was dumped. The prevalence of MDR UTI was higher in patients that used pit latrines, unprotected washing and drinking water, and did not always use soap for handwashing. As for socio-demographics, MDR UTI prevalence was higher if the household did not own a computer, owned their house, where neither the patient or the household head had secondary education, and in older (>45 years old) patients. MDR rates were higher if the household reported that they did not face obstacles accessing medicine, where a household member works in a hospital, and where they reported sharing antibiotics. Patients had higher MDR UTI prevalence if they reported having HIV/AIDS, have a disability, had surgery or been an inpatient within past 6 months, and did not know the term 'antibiotic'. Finally, if the patient had delayed seeking treatment for their UTI, attended a government clinic, took ABs for UTI, and had failed treatments, their MDR rates were higher.

                                                                                         

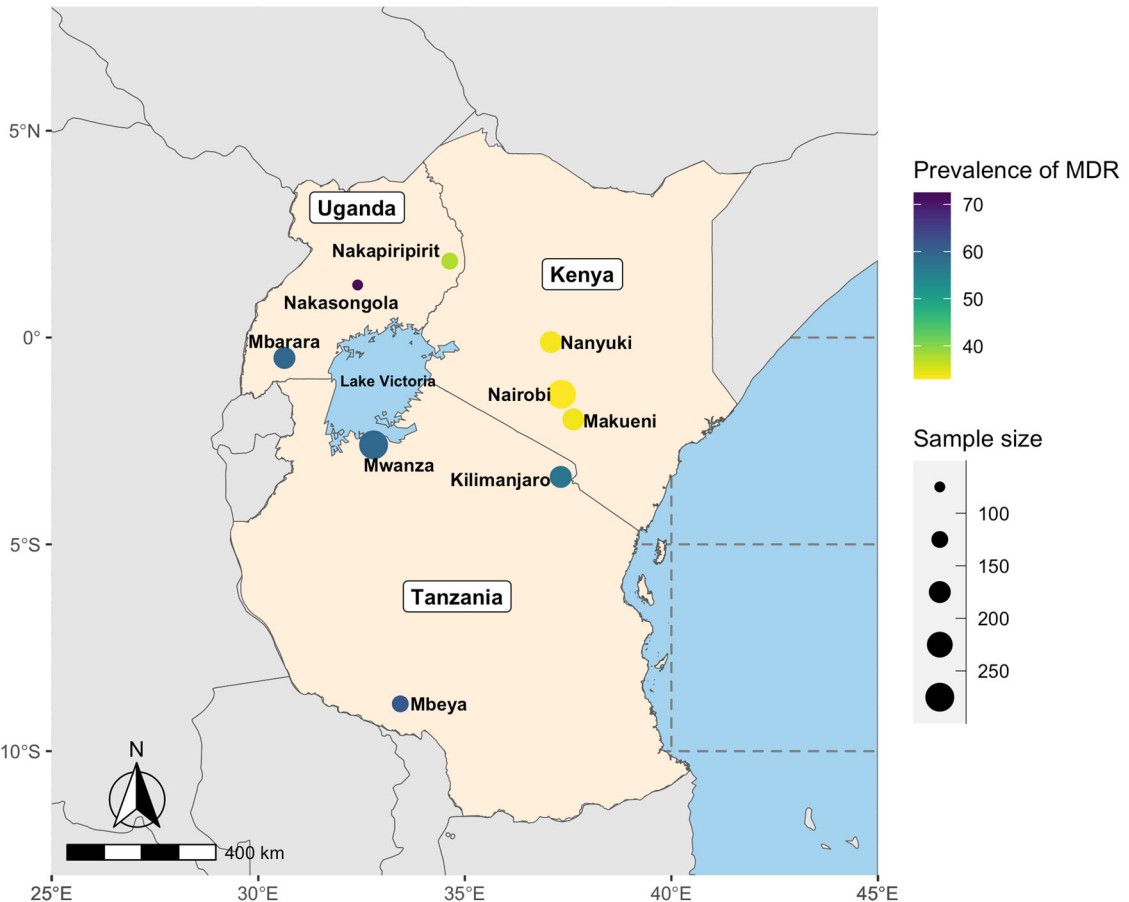

**Fig. 1 | Map of sites where data was collected in Kenya, Tanzania, and Uganda, their approximate sample sizes, and prevalence of MDR UTI.** The maps shows the 3 African countries where study sampling took place. Large bodies of water are shown in blue. The nine sampling areas are indicated with dots. The size of the dots corresponds to the absolute sample size within each site, that was used in our analysis, such that larger dots indicate a larger sample size. The colour of the dots corresponds to the percentage of MDR UTI found within each of the nine sites, with lighter yellow shades indicating a lower percentage of MDR, and darker blue shades indicating a higher percentage of MDR.

## Results of Bayesian profile regression

**Primary outcome: MDR UTI.** Bayesian profile regression generated 24 clusters ranging in size from 17 to 127 patients. Median risks of MDR UTI ranged from 0.26 to 0.72 across the clusters (Fig. S1). Overall 42 of the 67 variables were important for determining cluster allocation. In Fig. 3 we summarise how those factors jointly predict membership of the 10 high-risk clusters. The corresponding heat map for the 7 low-risk MDR UTI clusters is shown in Fig. S2. With many binary variables we often observe that risk factor associations are mirrored in low- and high-risk MDR UTI clusters. In Fig. 4 we list the factors with clear signals for an association with either low- or high-risk MDR UTI clusters. Detailed characteristics of the high-risk clusters, including 95% credible intervals for the cluster risks are shown in a series of figures available on github: https://github.com/katykeenan1981/hatuaprofilepaper/tree/main.

Contextual factors: As might be expected from the bivariate analyses, clusters with higher MDR UTI were more likely to contain patients from Tanzania or Uganda, and less likely from Kenya. Although international travel and urban and rural residence helped determine cluster membership, there was no clear signal for whether they were associated with high or low MDR UTI risk.

Household level factors: High-risk MDR UTI clusters were more likely to contain patients from households that kept animals, used ABs to raise livestock, that had contact with animal manure and were close to sites where people had dumped rubbish. Correspondingly, low-risk MDR UTI clusters had lower probability of containing patients with those characteristics. Having a household member who was sick was associated with higher-risk clusters, and not having one associated with lower-risk clusters. If people reported obstacles to getting medicines, they were more likely to belong to low-risk clusters, and vice versa.

Household WASH: Associations with drinking and washing water sources were not very clear. Half of the high-risk clusters (5 out of 10) were more likely to have protected private water sources, and correspondingly, many low-risk clusters had unprotected public sources. On the other hand, the high-risk clusters with the highest median levels of MDR UTI were more likely to have unprotected washing and drinking water. High-risk clusters were also more likely to have private flush toilets or pit latrines. Households that used soap when handwashing were more likely to belong to low-risk clusters.

Household socioeconomics: If the household head had lower education, a patient was more likely to belong to a high-risk MDR UTI cluster, and vice versa with higher education and lower-risk. Low-risk MDR clusters were more likely to have patients with health insurance and electricity. Patients in higher-risk clusters were more likely to own their own house.

Individual-level health and sociodemographic factors: Higher-risk clusters were more likely to contain patients aged over 45, and to have less than secondary education. Knowledge and familiarity with ABs was associated with risk but in different ways. Patients in higher-risk MDR clusters were more likely to recognise ABs from their packaging, but patients belonging to lower-risk MDR UTI were more likely to know the term 'antibiotic'. Patients in high-risk MDR UTI clusters reported more

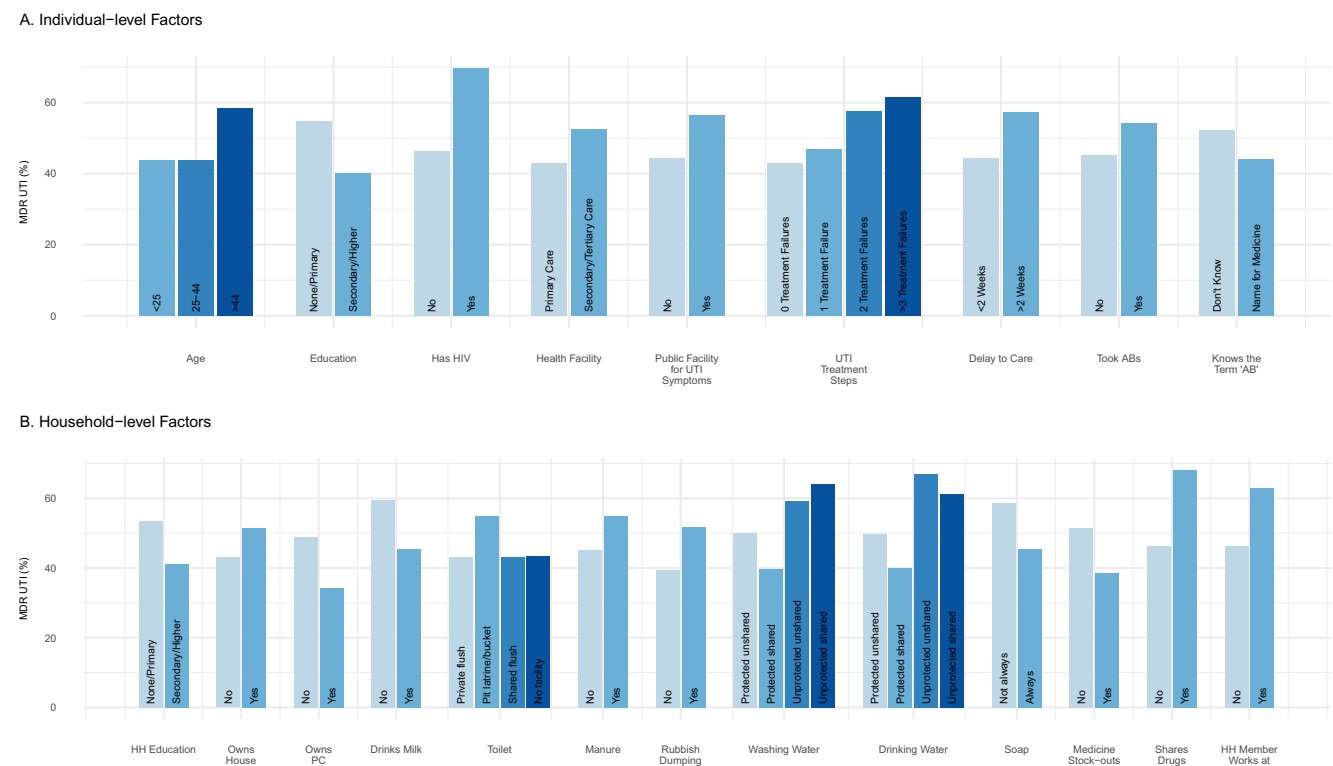

**Fig. 2 | Distribution of individual and household-level variables that have a significant association with MDR UTI.** Panel (**A**) shows individual level variables, panel (**B**) shows household-level variables. Statistical significance was assessed using two sided chi-square testing with false discovery rate (FDR) control[40]. Variables shown had FDR corrected *p*-values < 0.05. Exact *p* values are shown in Supplementary Data 2. Source data are provided as a Source data file.

treatment failures for UTI and more AB use as part of those treatments. Lower-risk patient clusters were less likely to delay UTI treatment seeking.

Individuals with all of the low risk profile characteristics in Fig. 4 were predicted to have a median MDR UTI proportion of 31%, versus 64% for those with all of the high risk profile characteristics. This varies slightly across sites but always shows at least a 20 percentage point difference (details in Fig. S3). The predicted disparity is lower when estimated using a priori defined multidimensional poverty[22] which only take account of education and living standards (Fig. S4).

**Sensitivity analysis**

When we used women only (*n* = 1369) (Figs. S5 and S6), 8 high-risk and 5 low-risk MDR UTI clusters emerged. Broadly the same patterns were observed, with some small differences. Having sick household members, which previously had been a clear signal for high-risk cluster, became less important, and having 4 or more live births emerged as a risk factor for MDR. The analysis when restricted to Gram-negative bacteria (*n* = 1009), also showed that largely the same variables emerged as important, which was expected as this represents the majority of the sample. The Gram-positive group was too small for a well-powered analysis.

When we repeated the analysis using the regionally specific ABR outcome, EA ABR, associations were again similar to the main analysis (Figs. S7 and S8). Profile regression generated 25 clusters where the average EA ABR median prevalence ranged from 11 to 88%. Four variables important for determining MDR UTI clusters were not important for EA ABR (work status, private clinics, treatment delay, and heard of ABR). Finally, we recategorized patients with intermediate AST results as being susceptible (rather than resistant), which reduced the proportion of patients with MDR UTI from 48% to 38% (Supplementary Data 1). The results using this outcome generated 25 clusters with 6

high-risk and 7 low-risk MDR clusters, fewer than in the main analysis. A similar number of variables were found to be important (41 variables vs 42) and the pattern of associations very similar (see Figs. S9 and S10).

## Discussion

In this study we investigated interrelations between a wide range of contextual, household, and individual risk factors for MDR UTI linked at the individual level. Many of these factors have been proposed as individual drivers of ABR and MDR development, transmission, or both. This study provides empirical evidence that over 40 environmental, social and economic factors are jointly associated with MDR UTI and each other, demonstrating how the co-occurrence of multiscalar risk factors produces intersectional inequality in ABR risk. This extends recent work which simply explored how treatment pathways correlate with MDR risk[36]. The findings are robust across different specifications of MDR UTI, and among subsamples of women and Gram-negative infections.

The application of Bayesian profile regression provides evidence for interrelationships both between the different risk factors and with ABR, that is not possible to observe from simple bivariate analysis or multivariate regression approaches[34–36]. This reveals clusters of characteristics that operate synergistically to shape unequal ABR burden. Socioeconomic deprivation clustered with sociodemographic, environmental and behavioural vulnerabilities and higher MDR risk. Higher MDR levels were found in older patients, those living in households with lower education levels, more ill health, and lower levels of material assets. These demographic and resource-related measures intersect with WASH vulnerabilities, represented by closer contact with animals and manure, waste dumping near the household, and less handwashing with soap. Occasionally there is some unexpected characteristic in the cluster profiles, for example house ownership being a risk factor for

| | Cluster-15 (N=73, Mdn=0.58) | Cluster-16 (N=93, Mdn=0.59) | Cluster-17 (N=71, Mdn=0.59) | Cluster-18 (N=63, Mdn=0.62) | Cluster-19 (N=55, Mdn=0.64) | Cluster-20 (N=76, Mdn=0.64) | Cluster-21 (N=56, Mdn=0.68) | Cluster-22 (N=35, Mdn=0.69) | Cluster-23 (N=68, Mdn=0.69) | Cluster-24 (N=18, Mdn=0.72) |
|---|---|---|---|---|---|---|---|---|---|---|
| International travel (yes vs no) | 0 | 0.16 | 0.04 | 0.13 | -0.11 | 0.1 | -0.09 | 0.23 | -0.13 | 0.76 |
| Tertiary/secondary vs primary care | 0.23 | 0.06 | 0.35 | -0.12 | 0.31 | -0.36 | -0.19 | -0.28 | -0.35 | -0.46 |
| Urban vs rural | 0.1 | 0.19 | 0.12 | 0.21 | -0.07 | -0.12 | 0.02 | -0.02 | -0.59 | -0.49 |
| Site: TZ: Mbeya | 0 | 0.43 | -0.02 | 0.07 | -0.06 | 0.1 | -0.05 | 0.23 | -0.07 | 0.63 |
| Site: TZ: Kilimanjaro | 0.56 | 0.08 | 0.65 | 0.05 | -0.08 | -0.02 | -0.08 | -0.04 | -0.08 | -0.07 |
| Site: TZ: Mwanza | -0.05 | 0 | -0.11 | 0.39 | -0.17 | 0.44 | -0.14 | 0.28 | -0.14 | -0.15 |
| Site: UG: Mbarara | -0.08 | -0.07 | -0.09 | -0.09 | 0.57 | -0.07 | 0.11 | -0.07 | 0.66 | -0.07 |
| Site: UG: Nakasongola | -0.04 | -0.04 | -0.04 | -0.04 | 0.13 | -0.03 | 0.56 | -0.02 | 0.05 | -0.02 |
| Site: UG: Nakapiripirit | -0.08 | -0.08 | -0.08 | -0.08 | -0.07 | -0.07 | -0.07 | -0.07 | -0.08 | -0.06 |
| Site: KY: Nairobi | -0.14 | -0.15 | -0.14 | -0.14 | -0.15 | -0.17 | -0.14 | -0.16 | -0.16 | -0.15 |
| Site: KY: Makueni | -0.1 | -0.1 | -0.1 | -0.1 | -0.1 | -0.1 | -0.1 | -0.1 | -0.1 | -0.09 |
| Site: KY: Nanyuki | -0.1 | -0.1 | -0.09 | -0.09 | -0.1 | -0.09 | -0.1 | -0.09 | -0.1 | -0.09 |
| Rubbish dumped near HH | 0.07 | 0.25 | 0.06 | 0.1 | 0.08 | 0.29 | 0.25 | 0.2 | 0.08 | 0.31 |
| Uses ABs with livestock | 0.21 | -0.07 | 0.22 | 0.11 | 0.14 | 0.25 | -0.08 | 0 | 0.26 | -0.07 |
| HH uses manure | 0.14 | -0.01 | 0.2 | 0.09 | -0.08 | 0.23 | -0.17 | 0.16 | 0.23 | 0.52 |
| HH has livestock | 0.36 | -0.03 | 0.46 | 0.13 | 0.16 | 0.38 | -0.22 | 0.15 | 0.28 | -0.29 |
| HH consumes milk | -0.08 | -0.13 | 0.03 | -0.15 | 0 | -0.12 | -0.06 | -0.19 | 0.07 | -0.47 |
| Drinking water: Unprotected private | -0.03 | -0.03 | -0.04 | -0.04 | -0.01 | -0.01 | 0.02 | 0.66 | 0.16 | 0.03 |
| Drinking water: Protected private | 0.39 | 0.47 | 0.45 | 0.4 | 0.29 | -0.3 | -0.1 | -0.3 | -0.34 | -0.11 |
| Drinking water: Unprotected public | -0.06 | -0.06 | -0.07 | -0.06 | 0.03 | 0.33 | 0.03 | -0.05 | 0.31 | 0 |
| Drinking water: Protected public | -0.31 | -0.39 | -0.34 | -0.31 | -0.31 | -0.03 | -0.03 | -0.34 | -0.13 | -0.03 |
| Washing water: Protected private | 0.4 | 0.49 | 0.48 | 0.41 | 0.29 | -0.29 | -0.07 | -0.31 | -0.35 | 0 |
| Washing water: Unprotected public | -0.04 | -0.04 | -0.05 | -0.05 | -0.03 | -0.01 | -0.03 | 0.68 | 0.16 | 0.19 |
| Washing water: Unprotected private | -0.07 | -0.07 | -0.09 | -0.08 | 0.04 | 0.35 | 0.05 | -0.04 | 0.38 | -0.06 |
| Washing water: protected public | -0.3 | -0.39 | -0.35 | -0.29 | -0.31 | -0.05 | 0.01 | -0.35 | -0.2 | -0.16 |
| Toilet: Pit latrine | 0.05 | -0.17 | -0.13 | -0.31 | 0.54 | 0.16 | 0.52 | 0.22 | 0.58 | 0.07 |
| Toilet: Private flush | 0.11 | 0.09 | 0.35 | 0.28 | -0.27 | -0.1 | -0.28 | -0.14 | -0.3 | -0.28 |
| Toilet: Public flush | -0.13 | 0.12 | -0.18 | 0.06 | -0.22 | -0.16 | -0.19 | -0.06 | -0.23 | 0.21 |
| Toilet: No facility | -0.04 | -0.05 | -0.05 | -0.05 | -0.05 | 0.09 | -0.05 | -0.04 | -0.05 | -0.03 |
| Raw sewage near HH | -0.01 | 0.03 | -0.03 | 0.24 | -0.07 | -0.06 | -0.01 | 0.33 | -0.18 | -0.17 |
| Uses soap when handwashing | -0.13 | -0.21 | 0.04 | -0.01 | -0.11 | -0.35 | -0.12 | -0.24 | -0.25 | -0.66 |
| HH owns house | 0.33 | -0.1 | 0.35 | 0.1 | 0.04 | 0.33 | -0.27 | 0.27 | 0.27 | 0.28 |
| HH owns land | 0.03 | -0.27 | 0.15 | -0.02 | 0.22 | 0.13 | -0.09 | 0.11 | 0.35 | -0.24 |
| HH has a smartphone | -0.03 | 0.02 | -0.02 | 0.36 | 0.09 | -0.11 | 0.16 | -0.08 | -0.09 | 0.11 |
| HH has electricity | -0.03 | 0.03 | 0.12 | 0.1 | 0.09 | -0.2 | -0.17 | -0.2 | -0.17 | -0.08 |
| Temporary or mud walls vs cement, brick or metal | -0.01 | -0.1 | -0.15 | -0.17 | -0.06 | 0.06 | -0.01 | 0.17 | 0.35 | 0.15 |
| HH has a TV | -0.06 | 0.06 | 0.26 | 0.38 | 0.02 | -0.36 | -0.25 | -0.4 | -0.34 | -0.06 |
| HH head has secondary+ education | -0.48 | -0.16 | -0.21 | 0.19 | -0.12 | -0.4 | -0.01 | -0.39 | -0.32 | -0.1 |
| Sick household member | 0.09 | 0 | 0.15 | 0.1 | 0.32 | 0.08 | 0.18 | 0.11 | 0.2 | -0.06 |
| Experienced AMR | -0.19 | -0.09 | -0.24 | -0.08 | 0.08 | 0.17 | 0.18 | 0.04 | 0.25 | -0.42 |
| Difficulty w/health costs | 0.18 | 0.02 | -0.37 | -0.4 | 0.29 | 0.05 | 0.22 | 0.13 | 0.22 | -0.51 |
| Lack of money is an obstacle | 0.45 | 0.19 | -0.24 | -0.25 | -0.16 | 0.28 | 0.19 | 0.17 | 0.19 | -0.32 |
| Has health insurance | -0.15 | -0.27 | 0.46 | 0.22 | -0.34 | -0.06 | -0.34 | -0.15 | -0.36 | 0.16 |
| Medicine stock outs are an obstacle | -0.26 | -0.28 | -0.14 | 0.02 | 0.02 | -0.21 | -0.13 | -0.23 | -0.19 | -0.28 |
| UTI treatment seeking: 1 treatment failure | 0.05 | 0.13 | -0.04 | 0.01 | 0.13 | -0.01 | 0.15 | -0.02 | 0.07 | -0.15 |
| UTI treatment seeking: 2 treatment failure | 0.15 | 0.02 | 0.1 | 0.11 | 0.08 | 0.09 | 0.14 | 0.19 | 0.18 | 0.16 |
| UTI treatment seeking: 3+ treatment failure | 0.16 | -0.03 | 0.22 | 0.11 | 0.08 | -0.04 | 0.09 | 0.11 | 0.18 | -0.05 |
| UTI treatment seeking: direct to clinic | -0.37 | -0.13 | -0.29 | -0.23 | -0.3 | -0.04 | -0.4 | -0.3 | -0.45 | 0.01 |
| Delayed seeking UTI treatment | 0.21 | 0.09 | 0.12 | 0.09 | 0.1 | 0.18 | 0.06 | 0.37 | 0.39 | 0.12 |
| Went to private clinic to treat UTI symptoms | 0.26 | 0.15 | 0.31 | 0.12 | 0.2 | 0.14 | 0.27 | 0.08 | 0.24 | 0.24 |
| Went to government clinic to treat UTI symptoms | 0.26 | 0 | 0.16 | 0.19 | 0.21 | 0.06 | 0.26 | 0.38 | 0.36 | -0.23 |
| Took ABs recently for UTI | 0.26 | -0.06 | 0.21 | 0.16 | 0.18 | -0.12 | 0.32 | 0.13 | 0.33 | -0.17 |
| Has recurrent UTI symptoms | 0.2 | -0.06 | 0.08 | -0.02 | 0.18 | -0.22 | 0.21 | 0.03 | 0.28 | -0.22 |
| Is familiar with ABs | -0.01 | 0.01 | -0.03 | 0.23 | 0.33 | -0.17 | 0.31 | -0.34 | 0.25 | -0.21 |
| Took ABs in past 6 months | 0.37 | 0.11 | 0.34 | 0.29 | -0.27 | -0.15 | -0.47 | 0.1 | -0.22 | 0.39 |
| Heard the term AMR | -0.06 | 0.1 | 0 | 0.04 | 0.16 | 0.02 | 0.22 | -0.12 | 0.18 | -0.28 |
| Patient difficulty with health costs | 0.2 | 0.01 | -0.49 | -0.33 | 0.04 | -0.04 | 0.08 | -0.05 | 0.19 | -0.22 |
| Feels UTI stigma | 0.06 | -0.14 | -0.05 | -0.15 | 0.28 | -0.18 | 0.3 | -0.13 | 0.39 | -0.18 |
| Knows the term 'antibiotic' | -0.28 | -0.36 | -0.07 | 0.12 | -0.09 | -0.47 | 0.24 | -0.43 | 0.25 | -0.32 |
| Age: 45+ years | 0.52 | -0.06 | 0.68 | 0.07 | -0.01 | 0.28 | -0.19 | 0.44 | 0.12 | 0.03 |
| Age: 25-44 years | -0.34 | -0.04 | -0.44 | 0.04 | -0.02 | -0.27 | 0.06 | -0.27 | -0.05 | -0.08 |
| Age: <25 years | -0.19 | 0.09 | -0.24 | -0.12 | -0.01 | -0.01 | 0.12 | -0.18 | -0.08 | 0.01 |
| Patient works | -0.06 | 0.07 | -0.17 | -0.01 | 0.2 | 0.14 | 0.13 | 0.07 | 0.33 | -0.45 |
| Has secondary educ or higher | -0.41 | -0.18 | -0.26 | 0.08 | -0.09 | -0.37 | 0 | -0.37 | -0.24 | -0.17 |

High risk MDR clusters (Cluster Size)

— — — — — — —  **Distance from the Median**  — — — — — — —

higher MDR. This could be because another risk factor, keeping livestock, reflects agricultural livelihoods and rural living which are more likely to also include owning land and housing as forms of wealth[37]. In accordance with previous work on AB misuse[22], we observe seemingly competing results for AB familiarity (confers higher MDR risk) and knowing the term 'antibiotic' (confers lower MDR risk). We point to our qualitative data which suggests the complexity of 'knowledge' and familiarity with ABs in this context[22]. Familiarity with medicines does not conflate with appropriate 'knowledge' about them or their use; familiarity with packaging may be due to greater infection risk and need for drugs, hence higher exposure. As might be expected, patients in high-risk MDR clusters

**Fig. 3 | Difference from the median values for 42 variables in the 10 high-risk MDR UTI clusters.** Figure 3 displays how the 42 important variables (y axis) are distributed within each high-risk MDR cluster (x axis). The variables are grouped thematically and within each theme, ranked according to the strength and direction of the associations with MDR. The numbers in the cells indicate the distance between the proportion of this characteristic in the whole sample and the median probability of having this characteristic in the specific cluster. The shading of the blue and red colours indicates the strength of the prevalence of the factor's category in the high-risk cluster, with deeper colours showing a higher prevalence. For example, a row which contains majority red blocks indicates that subjects that belong to a high-risk MDR UTI cluster are likely to have this factor characteristic, whereas majority blue blocks indicate that subjects that belong to a high-risk MDR cluster are not likely to have this factor characteristic. A mixture of blue or reds, or more neutral shades indicate no clear signal. For more detail, please consult the detailed PReMiuM plots in the supplementary materials. Source data are provided as a Source data file.

**Fig. 4 | Factors with clear signals for joint associations with low- or high-risk MDR UTI clusters, based on Bayesian profile analysis for MDR UTI.** The factors are identified from the list of 67 variables considered in the profile regression, which are described in the methods and in Table S3. The factors are ordered hierarchically according to the scale at which they were measured, starting from area-level, to community, to household, to individual-level. The red colour (right hand column of the chart) indicates factors with a clear signal for being associated with membership of high risk MDR UTI clusters. The blue colour (left-hand column) indicates factors with a clear signal for being associated with membership of low risk MDR UTI clusters.

reported more UTI treatment failures, including very recent AB use, indicating that MDR UTI is lengthening treatment journeys and further increasing healthcare costs and ongoing risk of infection and ABR for the poorer groups[21].

The burden of ABR is unequal[9,23], but so far there is limited empirical understanding of how socioeconomic dimensions operate to shape ABR evolution and transmission. In our sample, AB use behaviours and MDR burden do not correspond, speaking to further injustice. Lower education groups suffered the highest MDR burdens, despite being the least likely to misuse antibiotics[22]. Besides education, other dimensions contribute to differential risk of MDR (WASH, health insurance and access, underlying health status). These factors likely operate together to both promote ABR evolution and transmission for poorer groups. Our results suggest some mechanisms. First, contacts with animals and manure, sick family members, and deprived WASH, suggest increased exposure to bacteria and reservoirs of resistance[32]. Patient factors, such as older age and ill health increase vulnerability to (ABR) infections. Underfunded public healthcare, limited diagnostics and treatment options, may compound risk for poorer subgroups[21]. Fragmented systems of care, use of drug sellers where ABs are readily available in small doses without prescriptions could further promote the untargeted use or misuse of ABs which, in turn promotes ABR[21,22]. Finally, the care burden from MDR infections could further impact economic resources of households.

These results challenge behaviouralist accounts of ABR, which have focussed on intervening on antibiotic use and misuse as a solution to halt AMR. The fact that indicators of individual AB use over the last 6 months or AB misuse (poorer adherence or self-medication) were not clearly associated with higher risk of having an MDR UTI, supports the idea that AB use, while an important proximal driver, is not the only factor to address in the fight against ABR[33,35]. AB use operates jointly with other factors like older age, socioeconomic deprivation, unprotected water, and environmental waste exposure. This underlines the need to consider AB use behaviours as part of a broader social and material assemblage[22]. This study suggests that those designing interventions to optimise AB use should consider the wider social context of vulnerability and risk to be most effective.

The study has some limitations. The linked quantitative cross-sectional patient sample was representative of adults attending mainly public outpatient services with UTI-like symptoms, rather than the

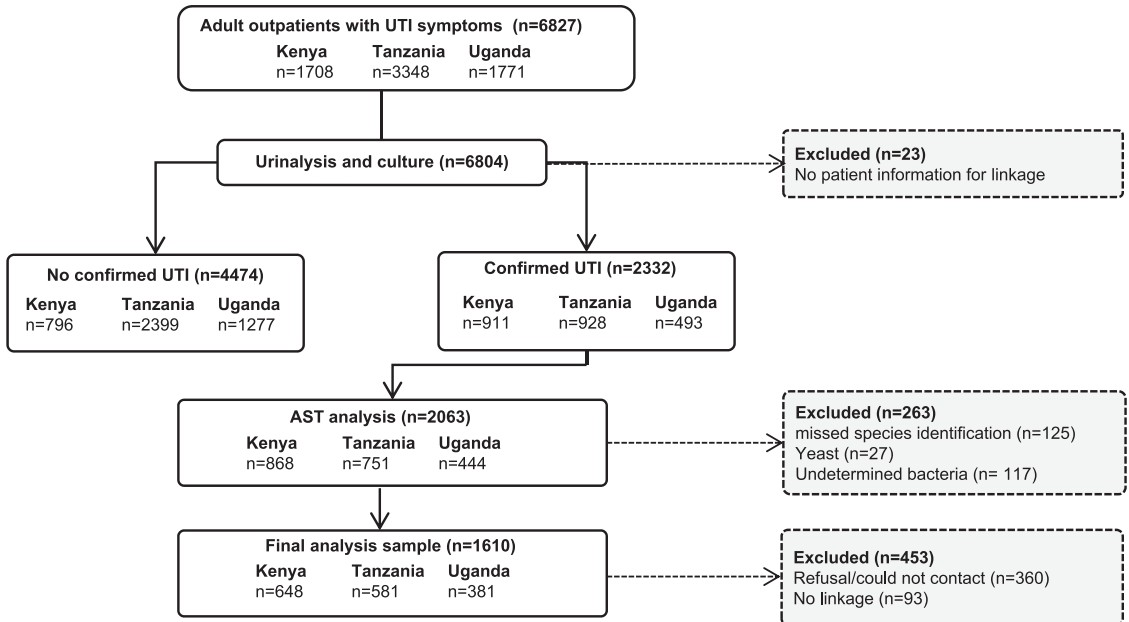

**Fig. 5 | Study recruitment and selection of the analysis sample.** The chart shows the recruitment flow and sample selection for the study. We recruited individuals with UTI symptoms attending clinics in our study areas. Following urinanalysis, we selected only those with confirmed UTI. Those patients' samples then underwent AST analysis to determine resistance within the uropathogen, and we attempted a follow-up visit to the household to collect further social economic and environmental data. The final sample for analysis includes those with valid AST linkage and a completed household visit.

general population or those with other infections (e.g., respiratory tract infections). The socio-demographic and behavioural variables we use are derived from self-report which may affect the associations observed. We observed strong MDR variations by site, a variable which likely proxies for many structural, political and economic factors[9,33] that we could not measure at sub-national levels. Kenya recruited a less diverse and relatively socioeconomically advantaged and urban sample, due to limitations imposed by the pandemic. It also contained proportionally more Gram-positive infections, which have higher resistance rates. However, although site contributes to cluster allocation, the findings was not driven by area-level differences. Sample size prohibited us from analysing MDR rates by sub-national sites, for men and by all bacterial species. There was a trade-off between breadth of sampling and ability to explore context-specific processes in detail; future studies could use our results to develop cost-effective mechanisms for collecting integrated surveillance data on variables shown to be important. Ongoing HATUA work will leverage genomic data to refine measurement of ABR and focus on better understanding the relationship to important variables identified here.

This study using linked One Health data and Bayesian profile regression reveals intersections between over 40 different ABR drivers in the environment, households and individuals. This demonstrates the complex interrelationship between environmental patterns and multidimensional poverty which shape vulnerability to infections and ABR. Further research is needed using detailed longitudinal data to understand directionality, to disentangle ABR development from transmission and to identify the most appropriate intervention points in specific settings.

## Methods
### Data and sample
The data were collected by the HATUA (Holistic Approach To Unravelling ABR) Consortium, an interdisciplinary, three-country study on the drivers of ABR, full details of which are described in the protocol[12]. The study used urinary tract infection (UTI), a commonly occurring bacterial infection, as a clinical prism through which to investigate the drivers of ABR, and collected data from patients in clinics, their

households and local communities. Sample selection and exclusions for the analysis are shown in Fig. 5.

Between February 2019 and September 2020, 6827 adult outpatients (aged 18 years and older, or those 14–18 years and pregnant, who comprised 1% of the sample) were recruited from several healthcare facilities in three countries (Kenya: Makueni, Nairobi, Nanyuki; Tanzania: Kilimanjaro, Mbeya and Mwanza, Uganda: Mbarara, Nakapiripirit, and Nakasongola). As per the protocol[12] sites were chosen to represent different levels of urbanisation, socioeconomic status and environmental exposures. Healthcare facilities were predominantly government-funded and included both primary, secondary, and tertiary levels of care in all countries (see Table S1 for details and recruitment dates). COVID-19 pandemic restrictions affected recruitment, having a higher impact in Kenya, where sampling took place over a shorter period and in fewer (higher level) facilities. In two Kenyan sites (Makueni and Nanyuki) we sampled only from secondary/tertiary hospitals, so proportionally fewer patients in Kenya are drawn from primary care. During face-to-face consultations, doctors or clinical officers identified patients with symptoms indicative of UTI for inclusion to the study. Less than 1% declined to participate, and this did not vary by site. Patients provided a mid-stream urine sample and answered a questionnaire on treatment-seeking, AB use, health factors, knowledge and attitudes around ABs, and socio-demographic characteristics ($n = 6804$). Among patients with microbiologically confirmed UTI (defined by the presence of >10$^4$ colony-forming units per millilitre (CFU/mL) of one or two uropathogens) and who consented to be recontacted, we conducted follow-up interviews in person in their homestead about a month later (mean days 31, IQR 5–42). At the household, a questionnaire was administered to the patient, or another adult member of their household, which covered household composition, socioeconomic factors, sanitation and hygiene, illness and health-seeking behaviour, and livestock practices ($n = 1610$). Interviewers also made observations of environmental features: sanitation, livestock, and hygiene practices. Each country had a minimum target sample size of 600 patients with microbiologically confirmed UTI. We linked questionnaire and microbiological data using anonymous patient identifiers. We compared the characteristics of patients

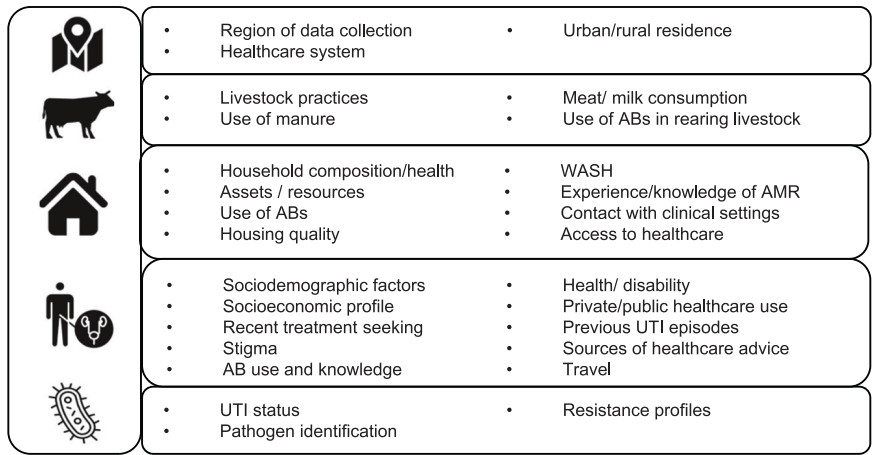

**Fig. 6 | Themes covered by the variables included in the analysis.** Summary of themes and the associated variables measured at various scales in this study. The themes are represented on the left-hand side with the associated variable grouped accordingly on the right-hand side. Themes are ordered hierarchically from top to bottom on their scale: Location (site; urban/rural residence), community, One Health and environment dimensions (livestock and farming practices, rubbish disposal), Household characteristics (WASH, household socioeconomic factors, health and AB use patterns), UTI patient individual factors (treatment seeking and AB use, health, socio-demographics, socioeconomic factors, health attitudes and knowledge) and finally microbiological data (bacterial identification and resistance profile).

---

with positive UTI cultures ($n = 2332$), those with antibiotic susceptibility testing (AST) ($n = 2063$) and those who were included in the analysis sample ($n = 1610$) (see supplementary Table S2), and these varied little.

### Ethical statement

Informed consent was obtained from all participants at the time of recruitment and at follow-up to the household. Pregnant women aged 14–18 also gave informed consent themselves in line with the ethical agreements in each country. Ethical approval for this project was obtained from the University of St Andrews, UK (No. MD14548, 10/09/19); National Institute for Medical Research, Tanzania (No. 2831, updated 26/07/19, CUHAS/BMC research ethics and review committee (No. CREC/266/2018, updated on 02/2019, Mbeya Medical Research and Ethics Committee (No. SZEC-2439/R. A/V.1/ 303030), Kilimanjaro Christian Medical College, Tanzania (No. 2293, updated 14/08/19). Uganda National Council for Science and Technology (number HS2406, 18/06/18); Makerere University, Uganda (number 514, 25/04/18); and Kenya Medical Research Institute (04/ 06/19, Scientific and Ethics Review Committee (SERU) number KEMRI/SERU/CMR/P00112/3865 V.1.2). For Uganda, administrative letters of support were obtained from the district health officers to allow the research to be conducted in the respective hospitals and health centres.

### Measurement of ABR and MDR

The patient urine samples underwent microbiological culture, pathogen identification and AST. Full methodological details and descriptive findings are reported elsewhere[13]. Our analytical sample comprises patients infected with Gram-positive ($n = 601$) and Gram-negative ($n = 1009$) bacteria. Gram-positive infections were more common in Kenya vs. Tanzania or Uganda[13]. Susceptibility to the tested ABs (see supplementary Table S1 for a full list of ABs tested) was determined by using the breakpoints (zone diameter interpretive criteria) indicated in the 2021 guidelines of the Clinical and Laboratory Standards Institute (CLSI)[38]. Our primary outcome was multidrug-resistant urinary tract infection (MDR UTI) (a binary classification) defined as urinary isolates resistant to at least one agent in three or more defined categories of antimicrobial agents, following the European Centre for Disease Prevention and Control (ECDC) guidelines[39]. We modified this to include nitrofurantoin and trimethoprim, two ABs routinely used for treating UTIs in this region

that are not included in the ECDC guidelines (Supplementary Table S3). In addition, for those species/genera not incorporated in the ECDC, i.e., *Salmonella* spp., *Shigella* spp., and *Streptococcus* spp., the MDR rates were calculated as above, but considering the resistance to a selected pool of tested ABs (Table S3). For the main analysis, isolates that showed intermediate resistance to a given AB were considered resistant, following local clinical practice. We also conducted a sensitivity analysis where we classified intermediate ASTs as susceptible. We constructed a secondary ABR outcome to reflect UTI treatment in East Africa (EA ABR). EA ABR was defined as resistance to any of the recommended ABs for treating uncomplicated upper and lower UTI from the National Treatment Guidelines (NTGs) for that country (for list see Table S4).

### Covariates: potential drivers of MDR

Using the questionnaire data collected at the healthcare facility and the household, the environmental observations made at the household, and geospatial data, we derived 67 variables which covered social, behavioural, clinical and environmental aspects potentially related to MDR UTI[12]. The question wording and coding are shown in Supplementary Data 1; some of these were explained and derived in earlier publications[13,21]. Figure 6 describes the linked dataset, with variables shown thematically and ordered from largest to smallest scale (location, community, household, patient, and pathogen).

### Statistical Methods

As a first exploratory step, we conducted bivariate analysis between all variables and the primary and secondary outcomes (MDR UTI, EA ABR UTI), using chi-square testing with false discovery rate (FDR) control[40]. We then used Bayesian profile regression[41,42], which clusters patients based on the variables described in Supplementary Data 1, and associates the patient clusters with high and low risk MDR UTI profiles. The underlying statistical procedure is described in detail in supplementary material, section 6. The method incorporates variable selection to indicate important variables for cluster allocation[42]. This allowed us to identify the most important variables from a large set of multiple interrelated factors, retaining the holistic nature of the data while looking for key patterns. We analysed the characteristics of patients within the high- and low-risk MDR UTI clusters, and used this to make inferences on how these combinations of characteristics contribute to differential risk of MDR. Missing responses for variables

were treated as unknown random quantities, imputed during the Markov chain Monte Carlo inferential procedure in the same manner to model parameters (maximum proportion of missing for any one variable was 3.4%). For those variables deemed important (selection probabilities greater than 0.69), we explored their bivariate correlations with each other and the outcome using Cramer's V with FDR control[40]. We also calculated average MDR UTI rates for predictive profiles based on standardised multidimensional poverty measures[22], and based on characteristics emerging from the profile regression. We used the fitted Profile regression model and the calcPredictions function of PReMiuM to do this.

We repeated the analysis considering the alternative outcome of EA ABR and the set of 67 covariates, with the addition of a variable measuring recent use of any of the ABs recommended for treating UTI in the relevant EA NTGs. Because literature suggest that UTI and ABR risks operate differently by gender[43,44], we also looked at women separately including pregnancy and parity as additional variables. The sample of men was too small for meaningful analysis. Finally, we stratified on bacterial species (Gram-positive vs -negative). Analyses and visualisations were performed with R version 4.3.2[45] and PReMiuM package version 3.2.13[46].

### Reporting summary

Further information on research design is available in the Nature Portfolio Reporting Summary linked to this article.

## Data availability

The data that support the findings of this study is available according to data sharing policy of the partners in the three participating countries, which restricts access due to ethical issues. The data forms part of a larger linked dataset, with ongoing analysis. To request access, please contact the Principal Investigator of the HATUA Consortium Professor Matthew Holden (mtgh@st-andrews.ac.uk) or the corresponding author (katherine.keenan@st-andrews.ac.uk). At the time of publication, further reuse of the data for analysis would require collaboration in the ongoing work of the HATUA Consortium. Source data for figures (where relevant) are provided with this paper. Source data are provided with this paper.

## Code availability

The analysis code is available through Github: https://github.com/katykeenan1981/hatuaprofilepaper/tree/main.

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

## Acknowledgements

We are grateful to the patients and communities who participated in the study. The Holistic Approach to Unravel Antibacterial Resistance in East Africa is a Global Context Consortia Award (MR/S004785/1, K.K., S.E.M., B.A., J.K., A.G.L., M.K., S.N., J.R.M., S.H.G., W.S., D.J.S., G.K., D.A., J.S., V.A.S., and M.T.G.H.) funded by the National Institute for Health Research, Medical Research Council, and the Department of Health and Social Care. This paper was funded in part by a grant from the National Institutes of Health (grant number U01CA207167, J.S.), and a Scottish Funding Council GCRF Consolidator Award (K.K., W.S.). The funders had no role in study design, data collection and analysis, decision to publish or preparation of the manuscript.

## Author contributions

K.K.: Conceptualisation, data curation, formal analysis, funding acquisition, investigation, methodology, supervision, project administration, visualisation, writing-original draft M.P.: Conceptualisation, formal analysis, investigation, methodology, software, supervision, validation, visualisation, writing- review and editing S.E.M.: data curation, funding acquisition, investigation, methodology, project administration, resources, supervision, writing- review and editing. B.A.: data curation, funding acquisition, investigation, methodology, project administration, resources, supervision, writing- review and editing. J.K.: data curation, funding acquisition, investigation, methodology, project administration, resources, supervision, writing- review and editing. A.G.L.: Conceptualisation, funding acquisition, supervision, writing- review and editing. M.K.: funding acquisition, writing- review and editing. S.N.: data curation, funding acquisition, investigation, methodology, project administration, resources, supervision, writing- review and editing. J.R.M.: data curation, funding acquisition, investigation, methodology, project administration, resources, supervision, writing- review and editing. M.F.M.: data curation, investigation, methodology, project administration, resources, supervision, writing-review and editing. W.J.: formal analysis, investigation, software, writing- review and editing D.L.G.: data curation, resources, writing-review and editing. E.O.: visualisation, writing- review and editing. Q.Z.: visualisation, writing- review and editing. R.S.: supervision, visualisation, writing- review and editing. K.J.F.: writing- review and editing. S.H.G.: funding acquisition, investigation, methodology, writing- review and editing. W.S.: funding acquisition, investigation, methodology, writing- review and editing. J.B.: data curation, investigation, methodology, project administration, resources, supervision, writing- review and editing. D.J.S.: funding acquisition, methodology, writing- review and editing. B.T.M.: funding acquisition, methodology, project administration, resources, supervision, writing- review and editing. G.K.: funding acquisition, methodology, writing- review and editing. D.A.: funding acquisition, methodology, writing- review and editing J.S.: funding acquisition, methodology, writing- review and editing V.A.S.: funding acquisition, methodology, writing- review and editing A.S.: data curation, methodology, project administration, writing- review and editing M.T.G.H.: funding acquisition, methodology, project administration, resources, supervision, writing- review and editing.

## Competing interests

The authors declare no competing interests.

## Additional information

**Supplementary information** The online version contains
supplementary material available at

Katherine Keenan.

**Peer review information** *Nature Communications* thanks Derek Cocker
and the other, anonymous, reviewer(s) for their contribution to the peer
review of this work. A peer review file is available.

[1]University of St Andrews, St Andrews, UK. [2]Catholic University of Health and Allied Sciences, Mwanza, Tanzania. [3]Makerere University, Kampala, Uganda.
[4]Kenya Medical Research Institute, Nairobi, Kenya. [5]Mbarara University of Science and Technology, Mbarara, Uganda. [6]Kilimanjaro Clinical Research Institute,
Kilimanjaro Christian Medical Centre, Moshi, Tanzania; Kilimanjaro Christian Medical University College, Moshi, Tanzania. [7]Africa Excellence Research Fund,
London, UK. [8]Oxford Big Data Institute, Oxford, UK. [9]Brigham and Women's Hospital, Boston, MA, USA. ✉e-mail: Katherine.keenan@st-andrews.ac.uk

## HATUA Consortium

David Aanensen[8], Annette Aduda[5], Benon Asiimwe ⓘ [3], Alison Elliott[10,11], Kathryn J. Fredricks[1], Stephen H. Gillespie ⓘ [1],
Dominique L. Green ⓘ [1], Matthew T. G. Holden ⓘ [1], Catherine Kansiime[3], Katherine Keenan ⓘ [1]✉, Mike Kesby[1],
Gibson Kibiki[7], John Kiiru[4], Andy G. Lynch ⓘ [1], John Maina[5], Blandina T. Mmbaga ⓘ [6], Stephen E. Mshana[2], Martha F. Mushi[2],
Joseph R. Mwanga[2], Stella Neema[3], Wilber Sabiiti[1], Alison Sandeman ⓘ [1], Derek J. Sloan[1], V. Anne Smith ⓘ [1] & John Stelling[9]

[10]London School of Hygiene and Tropical Medicine, London, UK. [11]MRC/UVRI & LSHTM Uganda Research Unit, Entebbe, Uganda.

