## [Transparent Peer Review file · Nature Communications]

Intersecting social and environmental determinants of multidrug-resistant urinary tract infections in East Africa beyond antibiotic use

Corresponding Author: Dr Katherine Keenan

Version 0:

Reviewer comments:

Reviewer #1

(Remarks to the Author)

Thank you for the opportunity to review this interesting paper.

The authors have examined data from outpatients diagnosed with UTI from a range of healthcare centres at nine sites within three East African countries, Kenya, Uganda and Tanzania.

The authors then analysed characteristics of patients with multi-drug resistant UTI compared to non-resistant UTI infections. The multivariable analysis is carried out using Bayesian profile regression.

A strength of the study is the large sample and associated variables collected on each case. There are some aspects of the paper, however, that would benefit from some further exploration or explanation.

Specific comments

Introduction

Line 40: The introduction states that the authors address existing gaps by 'using an appropriate study design', what is the study design of this paper and why is this considered to be 'appropriate'? It would be helpful to state this explicitly.

Introduction – it would seem relevant to include literature on risk factors for MDR UTI (and/or non-MDR UTI) either in African countries or more broadly as part of the introduction.

Lines 68-82, this section reviews papers that have examined drivers of colonisation with antibiotic resistant bacteria (ABR), and takes this approach forward to the present study. However, the present study is looking at MDR infections versus non-MDR infections in outpatient settings, which is potentially different to colonisation. What evidence is there that the same risk factors/models apply for MDR UTI infection as for human colonisation with non-pathogenic ABR organisms? Clarifying whether there is evidence that these are linked would be a useful addition.

A framework of the causal pathways to UTI versus MDR UTI may help provide a clearer understanding of why the authors chose to include so many variables in the analysis, potentially with hypothesis framing as well to explain the approach to the analysis. Use of DAG models, or a systems level map might also help to frame the analysis.

Methods

More details could be provided in the recruitment and study design.

The types of healthcare facilities used for recruitment seemed to vary considerably by country. The patient population in Kenya is clearly different to that of the other two countries, with 53% of the patients with UTI symptoms having a positive UTI (911 out of 1708) in Kenya, compared to 28% in Uganda and 28% in Tanzania. Why was the percentage of confirmed UTI so high in Kenya? Kenya also had more secondary/tertiary recruitment centres and much higher educational attainment of patients/participants. What was the rationale for the urban/rural split, what was the target sample size for recruitment by country or healthcare centre. Why was the sample size for Tanzania so much larger than the other countries?

Table S1 could be improved, currently this provides a number/code and a hyperlink to policy documents which means having to look up the relevant information for each country. The link to the Uganda and Tanzania documents did not work when I tried this. Listing the type of facility in each site would be simpler.

How did the study ensure that participants were only included once during the time period of the study? Or did the study design allow repeat observations for the same patient/participant to be included? Was the period of recruitment exactly the same for every study site?

Line 130-131: Can a rationale/supporting references be provided for categorising all intermediate ASTs as resistant.

Analysis/results

Could the authors state whether pregnancy was recorded (at the time of diagnosis) or parity for women. These would seem important variables to include.

The analysis considers UTI as a homogenous outcome and I wondered whether this is supported by literature. Are risk factors for UTI the same for men and women? Did the authors consider analysing the outcome in females separate to males? The number of cases in males is probably too small to examine alone, but would the results be any different if the analysis was on females only.

Similarly, a first presentation of UTI versus a multiple presentation/recurring UTI could also be conceived as possible source of heterogeneity in the outcome – was this explored at all?

The profile regression is an interesting approach, but also makes results quite difficult to interpret. It also makes it potentially difficult to draw any comparisons with other study findings (for example, should researchers want to extract data for systematic review and meta-analysis).

Could the authors explain why more typical adjusted/unadjusted regression models were not considered appropriate (using the factors identified in high clusters perhaps?).

Discussion

Lines 293 – 296 –it would be good to acknowledge that these are self report data therefore a lack of association could be either due to the nature of self-report data, or it could be that these factors are not associated with the outcome.

The limitations of the study could be recognised more fully regarding the sampling strategy, the cross-sectional nature of the study, the reliance on self-report for some of the antibiotic use data and behavioural factors.

Some demonstration of the robustness of data and methods would strengthen the discussion.

Reviewer #2

(Remarks to the Author)

This is a well written, highly relevant article, which highlights the importance of social disadvantages and One Health factors associated with AMR in East Africa. Their focus on co-occurrence of risk factors, rather than competing or individualised risks provides an interesting insight. Furthermore, their data illustrating that antibiotic use is only one of many drivers of AMR and that a wider social contextualisation is required, is a much-needed statement.

The suggestions made are to strengthen the authors points and support for their claims and to frame this work better within the context of available data.

Line 72-82: It is nice to see that other pertinent studies from East Africa have been highlighted. However, the authors state these are small, interfering that this study is somewhat larger. Of note, the studies mentioned provided more microbiological sampling than this study, and therefore it could be argued that is study was smaller in comparison. As stated in line 304, the sampling size of this study was too small to determine ABR rates by site or bacterial species, which would have been very useful; markedly improving the impact of the results. I would, therefore, not recommend focussing on comparison of size, but on comparison between approaches and results, highlighting the consistencies and differences in what they found/represent.

Line 72-82: The authors then provide further comparison, stating that they standardise their results across more sites and countries and therefore are not limited by geography. Again, I would say that all these studies, including this one, are limited in their scope and geographical spread. This study does not address these geographic gaps, but simply provides a different sampling frame. I think this paragraph & the discussion would be improved if it the authors would focus on the limitations across One Health datasets, and the absence of standardised approaches or large-scale analysis that precludes comparison and prioritisation. They then might consider which approaches or minimum datasets would be required to enable better risk evaluation or future analysis in the discussion.

Line 130: Was analysis completed on whether re-classifying intermediate to sensitive (rather than resistant) undertaken, and if so, did this markedly changed the results?

Lines 144-163 & 270: Here, I am unable to comment on whether the Bayesian approaches taken can show that the results “operate synergistically to shape ABR burden” (as they state in line 270), or whether in fact, other interpretations for these

results could be inferred. I would appreciate it if further statistical opinion was sought as to whether this claim is entirely accurate.

Line 170: Kenya seems to have less ABR than the other countries. Would the authors like to state later on, why they think this is the case, and whether these data / differences correlate to other available data from these settings?

Line 182 and Figure 4: Did HIV individuals have co-trimoxazole prophylaxis, which could have impacted on their results, and was this accounted for in 56% taking antibiotics? I also note that MDR UTI was higher in HIV (line 201). Again, did the role of CPT have any effect?

Line 236-239: Owning your own house was associated with a high-risk cluster, which might be argued serves as a proxy for higher SES, and contrary to the other information. Why do the authors think this result occurs and if this is contrary to other information, does it highlight a limitation of grouping results together?

Line 242-244: Again, recognising AB packaging was associated with a high-risk cluster, whereas recognising the word antibiotic was associated with lower risk cluster. How do the authors justify these somewhat competing results?

Lines 251: Interesting that EA ABR were similar, as was the analysis specifically on gram negative bacteria. Are we to assume then that these broad results are likely to represent those for gram-negative UTIs? If so, this should be stated.

Line 260: "Unprecedented"? As per my previous comments, this is merely a different analysis and sampling frame. Please remove.

Line 268-300: The importance of WASH and environmental factors were clearly stated in the introduction, yet this message seems a little lost in the discussion. For example, animals, manure and waste dumping have higher ABR risks, and low risk clusters were associated with handwashing with soap. Whilst the authors mention "toilet features", "hygiene", "unprotected water" and "environmental waste" within the discussion, this could be written more cohesively to make it clearer to the reader that these variables are environmental/WASH health proxies, and that poverty and paucity of WASH are part of an interlinked / intersectional vulnerability.

Line 275-277: Nice to see it highlighted that ABR UTIs could provide a cycle of costs for the poorest / most vulnerable people. Consider adding whether this might, in fact, not just increase their costs, but also increase their ongoing risks through further healthcare exposures and AB exposures. Breaking these cycles is likely to have many impacts for these high-risk individuals.

Line 284-291: Overall a good section, but the authors might like to consider how specific pathways or delivery of healthcare systems could place vulnerable individuals at increased risks too, and consider adding this in.

Line 291: ? "care burden patients". Should this read care burden from ABR infections?

Line 292-300: Excellent.

Line 305 (point 1): It is a shame that this study couldn't analyse results by site and in particular, MDR species, as this would have allowed a more refined understanding of the key risks. This highlights once again the differences between this, compared with other studies, and an overarching issue with One Health data. That being, the balance between the level of sampling/ recruitment to enable granular yet affordable results. This enhanced data is often unfeasible in low-resource settings, due to costs, technical constraints or geographic inequities. Furthermore, One Health data is frequently skewed towards human health, prohibiting the integration with other datasets. I would very much appreciate if the authors could state more clearly the nuances of this limitation and consider the context of East Africa / One health. They might also want to briefly postulate any feasible approaches that could be considered to address this?

Line 305 (point 2): Consider adding in whether you can or cannot infer any directionality, especially around how poverty affects ABR risks.

Line 305 (point 3): Another limitation is that some information obtained is from reported data, which is subject to bias, particularly around WASH behaviours (i.e. handwashing with soap).

Line 305 (point 4): Some aspects (i.e. WASH) have sparser questions (Table s5), providing only high-level data. It would be good to see how and why variables were chosen. Was this done using previously validated datasets or at the researcher's discretion / consortium consensus? Also, was this refinement done as part of the pilot period between 2017-18 (in protocol).

Line 312: "is only facilitating". Consider change to "is only associated with" for clarity to the reader.

Discussion (general): Is there a plan to use genomic information to enhance or refine these results? Additionally, is further modelling planned within HATUA? If so, these would be important things to state.

Data (general): I found the preceding protocol paper and supplementary data highlighting the statistical methods very useful, and allows for open reproducibility. However, is there a reason why the primary (anonymised) data is not accessible via a secure repository, to enable further analysis / meta-analysis to be undertaken? This would be of wider benefit to the AMR

community and enable cross country comparison with other datasets.

Line 449. Please use “One Health” or “one Health” throughout for consistency.

Figure 4: HH members work at hospitals have higher risk. Would the authors like to comment more on the role of healthcare exposure?

Supplementary data. Still has tracked changes in (for example S3I).

Reviewer #3

(Remarks to the Author)

• What are the noteworthy results?

Authors present data from a robust sampling frame including three different east African countries. Results included microbiology, patient, household and community level data linking the primary outcome, UTI ABR infections with 67 variables. Bivariate analyses and Bayesian profile regression were used to determine the joint association of these factors with ABR UTIs. Findings are presented in terms of high risk and low risk clusters for UTIs. Characteristics of high risk clusters were related to level of education, more chronic illness, resource-deprived households, contact with animals, and human/animal waste, not antibiotic use.

• Will the work be of significance to the field and related fields? How does it compare to the established literature? If the work is not original, please provide relevant references.

This work is of high significance and will contribute to describe risk factors, as high and low risk clusters, to antibiotic resistant UTIs. One of the major strengths of the paper was that 1610 individuals with microbiologically-confirmed UTI were included providing a robust sampling frame appropriate for the analyses carried out. Factors related to socioeconomic deprivation clustered with sociodemographic environmental and behavioural vulnerabilities had a higher ABR risk. Importantly, antibiotic use was not a risk.

• Does the work support the conclusions and claims, or is additional evidence needed?

Yes

• Are there any flaws in the data analysis, interpretation and conclusions? Do these prohibit publication or require revision?

No

• Is the methodology sound? Does the work meet the expected standards in your field?

Yes

• Is there enough detail provided in the methods for the work to be reproduced?

Yes

1) The use of the term “integrated” is vague, and authors should specify what is meant or implied by

Line 55: However, the contribution of these factors alongside antibiotic use is rarely investigated in an integrated way.

2) There are some results that appear to be presented in the methods section and should be considered for moving to the results section.

Line 98 Less than 1% declined to participate, and this did not vary by site.

Line 109 We compared the characteristics of patients with positive UTI cultures (n=2,332), those with antibiotic susceptibility testing (AST) (n=2,063) and those who were included in the analysis sample (n=1,610) (see supplementary Table S2), and these varied little.

3) Its unclear if the following is a previously published data point or a result of the study (in the methods section). If it is a result of the current analysis, consider moving to the results section.

Line 119: Gram-positive infections were more common in Kenya (48%) vs. Tanzania or Uganda (36% and 22% respectively). Susceptibility to the tested Abs...

4) The importance of the assessment of antibiotic use in combination with all of the other variables in question was emphasized in the introduction. In the results, however, there was little mention of antibiotic use results. Consider expanding the amount of information given for this point in the results section, in addition to the text below.

Line 179: Treatment seeking behaviours for UTI (i.e., number of steps in seeking treatment, self-treatment, AB use, clinic attendance) was generally more complex in Tanzania and Ugandan patients than those in Kenya.

5) Consider improving the clarity of the following sentence by specifying what obstacles were observed (or measured) and what the results of those observations were.

Line 199: MDR rates were higher if the household did not face obstacles accessing medicine, where a household member works in a hospital, and where they reported sharing antibiotics.

6) The following sentence is confusing and should be clarified

Line 279: In our sample, behaviours and burden do not correspond, speaking to further injustice.

7) There is a double period at the end of the sentence in line 286 that should be edited.

Reviewer #4

(Remarks to the Author)

Summary:

This paper deals with the antibacterial resistance (ABR) and investigates the joint contribution of behavioral, environmental, socioeconomic, and demographic factors which drive higher ABR risk. The data come from healthcare facilities in Kenya, Tanzania, and Uganda. The primary outcome was multi-drug resistant urinary tract infection (MDR UTI). The authors used Bayesian profile regression (BPR) where 67 variables based on individual, household and community-level features were used to develop clusters and relate these clusters to outcomes. Results suggest that individuals that high-risk MDR clusters were more likely to have disadvantages such as being old, lower education, more chronic illness, lived in resource-deprived household, etc.

Overall, the paper is well written and provides a welcome relief from tired old analyses that examine effects one variable at a time where the effects of one variable are dependent on fixed levels of other covariates. (Note this is the case even if interactions are included in the standard regression model; The effect of one variable may change depending on the *fixed* level of another variable, but such standard models do not examine how such explanatory variables covary jointly.)

Major issues:

Despite the advantages of the approach used here, the results are still a bit challenging to distill, even with the helpful “heat map” provided. A wonderful addition to this paper would be to provide a joint (not just individual) comparison of predictive profiles regarding their effect on MDR UTI. For example, one could examine “typical” privileged, middle-class, and underprivileged covariate profiles, re and compare their risks using the predictive profile features of the PReMiuM R-based package. These typical profiles must be defined a priori, from the literature or from the current data. Note that stitching together marginal averages of covariates may be inconsistent with the joint nature of the profile regression approach.

Minor Issues:

The outcome is MDR UTI – multi-drug resistant urinary tract infection – 3 or more categories of antibiotics. Did the authors utilize a sensitivity analysis using, say, 4 or more categories?

“AB’s” (line 49) – should this be defined?

“leap forward” – line 78 – ambiguous - reword

Line 85: “The data were ‘generated’”. This suggest that the data were synthetic. Is this what the authors mean?

Line 149: Which type of variable selection (binary or continuous)?

How was a variable deemed to be important?

Line 156: How much missing?

Line 162: Rstudio is really just a front end to R, so please give the version of R and the version of PReMiuM used.

Version 1:

Reviewer comments:

Reviewer #1

(Remarks to the Author)

The authors have addressed comments, increased the transparency of reporting for the sampling and participant recruitment process and made helpful clarifications to the methods and underlying rationale. The addition of sensitivity analyses where antibiotic susceptibility tests in the intermediate category are coded as sensitive rather than resistant, and a separate analysis for women only, is appreciated.

Overall the work sheds valuable light on the contexts surrounding multi-drug resistant urinary tract infections in the selected study areas.

Reviewer #2

(Remarks to the Author)

The authors have addressed the comments raised within the revised manuscript and supplementary materials.

It was pleasing to see a sensitively analysis was undertaken on intermediate ASTs being classified both as susceptible and

resistant, and the impact this had on results. The discussion has been improved by a more consistent messaging, alongside recognition of unexpected or conflicting clustering results and an expansion of the study limitations.

Reviewer #4

(Remarks to the Author)

After carefully reading the authors responses to my queries, I believe that my concerns have been adequately addressed.

Response to reviewer's comments

Note: comments are in bulleted italics, our responses are in normal text.

Reviewer #1 (Remarks to the Author):

- *Introduction Line 40: The introduction states that the authors address existing gaps by 'using an appropriate study design', what is the study design of this paper and why is this considered to be 'appropriate' ? It would be helpful to state this explicitly.*

We agree that this is ambiguous and have edited this to read "In this study, we address these gaps using a study design and methods which capture how diverse ABR drivers operate synergistically". Further on in the introduction, we also expand on how the method employed in our study is appropriate to study this system of ABR (partially in response reviewer 2's comments): "These previous studies predominantly rely on regression methods that obscure complex interrelationships between variables. In this study, we address these gaps by employing standardised data across several East African sites and interrogating these using statistical methods capturing interrelationships between risk factors and the outcome"

- *Introduction – it would seem relevant to include literature on risk factors for MDR UTI (and/or non-MDR UTI) either in African countries or more broadly as part of the introduction.*

We have added this.

- *Lines 68-82, this section reviews papers that have examined drivers of colonisation with antibiotic resistant bacteria (ABR), and takes this approach forward to the present study. However, the present study is looking at MDR infections versus non-MDR infections in outpatient settings, which is potentially different to colonisation. What evidence is there that the same risk factors/models apply for MDR UTI infection as for human colonisation with non-pathogenic ABR organisms? Clarifying whether there is evidence that these are linked would be a useful addition.*

We have edited this section to address this comment and those of the editor. In the new version, any studies which discuss only colonization (without mention of pathogenic infection) have been removed. We also include studies which address risk factors for MDR UTI specifically, although these are much fewer. Given that MDR is a simplified binary way of measuring ABR, we would suggest it is appropriate to draw on the ABR UTI literature as well, so some of the references were retained.

- *A framework of the causal pathways to UTI versus MDR UTI may help provide a clearer understanding of why the authors chose to include so many variables in the analysis, potentially with hypothesis framing as well to explain the approach to the analysis. Use of DAG models, or a systems level map might also help to frame the analysis.*

We have previously developed a theoretical framework based on the notion of assemblage, which was informed by literature review/ pilot study. This guided the design of the study, including the range of variables included, and analysis approach. This is described in the protocol ². We now refer to this in the introduction to orientate the reader:

“The study builds on evidence-based assemblage framework hypothesising how the drivers of UTI and ABR are interrelated (Asiimwe et al. 2021)”

- *Methods: More details could be provided in the recruitment and study design. The types of healthcare facilities used for recruitment seemed to vary considerably by country. The patient population in Kenya is clearly different to that of the other two countries, with 53% of the patients with UTI symptoms having a positive UTI (911 out of 1708) in Kenya, compared to 28% in Uganda and 28% in Tanzania. Why was the percentage of confirmed UTI so high in Kenya? Kenya also had more secondary/tertiary recruitment centres and much higher educational attainment of patients/participants. What was the rationale for the urban/rural split, what was the target sample size for recruitment by country or healthcare centre. Why was the sample size for Tanzania so much larger than the other countries?*

We have added more details on rationale for choice of sites to the methods section (“As per the protocol²⁵ sites were chosen to represent different levels of urbanisation, socioeconomic status and environmental exposures”). In Kenya, we had less diversity of recruitment facilities than the other countries and sampled from proportionally fewer primary facilities. This was mainly due to COVID-19 restrictions which made it difficult to work in primary healthcare facilities and to travel around. We used the same recruitment protocols but the communities from which they were drawn are less diverse in Kenya (concentrated in and around the capital). Each country had a target sample size of 600 UTI positive samples (details on this also added to the manuscript), but Tanzania chose to keep sampling after reaching this total. This country was less affected by pandemic restrictions than Uganda or Kenya which had strict travel bans in place.

- *Table S1 could be improved, currently this provides a number/code and a hyperlink to policy documents which means having to look up the relevant information for each country. The link to the Uganda and Tanzania documents did not work when I tried this. Listing the type of facility in each site would be simpler.*

We have updated this table to more clearly describe the types of facilities that are considered primary, secondary and tertiary in each country (with working, stable links). We have also updated Table S1 to show for each site whether we have sampled from primary, secondary and tertiary facilities and the dates of recruitment.

- *How did the study ensure that participants were only included once during the time period of the study? Or did the study design allow repeat observations for the same patient/participant to be included? Was the period of recruitment exactly the same for every study site?*

Repeat sampling was not allowed and prevented by a screening process at recruitment stage in the clinic (names were collected, and the process was run by fieldworkers familiar with the community setting). The period of recruitment for each site varied (see details added to Table S1). Some sites experienced delays in starting recruitment due to hold ups in ethical clearances (e.g. some sites in Kenya) and this was further complicated by the COVID-19 pandemic which restricted travel, so that sites in Kenya (Makueni and Nanyuki) had a shorter and more intense period of recruitment than in some other sites.

- *Line 130-131: Can a rationale/supporting references be provided for categorising all intermediate ASTs as resistant.*

We recognise that the category of Intermediate is subject to some debate as to how best to treat it in clinical practice (e.g. <https://clsi.org/about/blog/re-exploring-the-intermediate-interpretive-category/>). As acknowledged in this commentary, intermediate resistance has been seen as a surrogate for resistant and often leading clinicians to look for an agent for which the result was sensitive³. Whilst EUCAST have now adopted recommendations for "intermediate" (I) to be redefined as "Susceptible, increased exposure" (https://www.eucast.org/fileadmin/src/media/PDFs/EUCAST_files/Guidance_documents/To_clinical_colleagues_on_recent_changes_in_clinical_microbiology_susceptibility_reports_9_July2021.pdf), however this has not yet been aligned with CLSI guidelines. We therefore took a pragmatic approach, led by local clinical practice and guided by clinical advisors within the HATUA team, of considering Intermediate as another category of "non-susceptible" alongside Resistant.

However, the reviewer's question prompted us to explore the impact of categorising all intermediate ASTs as susceptible in a sensitivity analysis described in the methods and results, and supplementary materials (in particular Figs S8 and S9). The impact is to shift some patients who were previously classified as MDR=1 to MDR=0 reducing the MDR sample proportion from 47.6% to 37.7% (see Table S5), and in the profile regression analysis, this reduces the number of high-risk MDR clusters we detect from 10 to 6. However largely the same variables emerge as important for determining the high-risk clusters. Therefore, we would argue that our results are robust, however intermediate ASTs are treated. We have updated the text in the manuscript to provide more information about the categorisation the additional analysis done.

- *Analysis/results: Could the authors state whether pregnancy was recorded (at the time of diagnosis) or parity for women. These would seem important variables to include.*

We did collect pregnancy status and number of live births for women. Previously we excluded these because they were missing for men. However now we run a women-only model as a sensitivity analysis and we descriptively report these variables in Tables S5 and S7. For the women-only analysis these variables are included as variables determining the clustering into high and low-risk MDR clusters.

- *The analysis considers UTI as a homogenous outcome and I wondered whether this is supported by literature. Are risk factors for UTI the same for men and women? Did the authors consider analysing the outcome in females separate to males? The number of cases in males is probably too small to examine alone, but would the results be any different if the analysis was on females only.*

Thank you for this suggestion. We now include some literature which discusses how the risk of UTIs and ABR/MDR varies by gender (cited in the introduction and methods). We also conduct a women-only analysis, and this is reported in the results and supplementary materials (Figs S4 and S5)

- *Similarly, a first presentation of UTI versus a multiple presentation/recurring UTI could also be conceived as possible source of heterogeneity in the outcome – was this explored at all?*

We did use a variable 'Has recurrent UTI symptoms' which captures multiple presentation - the patient self-reporting previous presentation with UTI symptoms to the doctor in addition to the current episode. This was not associated with MDR UTI in simple bivariate testing, but does contribute to the formation of high-risk clusters (shown in Figure 5). It was a moderately

strong factor for belonging to 3 out of 10 high-risk MDR clusters, but in some clusters had the opposite or no effect- hence we described this as 'no clear signal' and it was not included in the infographic Fig 6.

- *The profile regression is an interesting approach, but also makes results quite difficult to interpret. It also makes it potentially difficult to draw any comparisons with other study findings (for example, should researchers want to extract data for systematic review and meta-analysis). Could the authors explain why more typical adjusted/unadjusted regression models were not considered appropriate (using the factors identified in high clusters perhaps?).*

As we explain in the introduction, we decided not to use ordinary regression-based approaches because they suffer from low power, especially when interaction terms are included, plus problems resulting from collinearity. This obscures complex variable inter-relations important for understanding ABR. To include this would be a different kind of paper. Now that we have identified a smaller set of important variables with clear signals, we will pursue more directed approaches in other papers. We now mention this in the discussion.

- *Discussion: Lines 293 – 296 –it would be good to acknowledge that these are self report data therefore a lack of association could be either due to the nature of self-report data, or it could be that these factors are not associated with the outcome.*

We now acknowledge this (“As these are self-reported variables response bias could be an alternative explanation”).

- *The limitations of the study could be recognised more fully regarding the sampling strategy, the cross-sectional nature of the study, the reliance on self-report for some of the antibiotic use data and behavioural factors.*

We have expanded the limitations to cover these points.

- *Some demonstration of the robustness of data and methods would strengthen the discussion.*

We have added some words to this effect (“*The findings are robust across different specifications of MDR UTI, and among subsamples of women and Gram-negative infections.*”).

Reviewer #2 (Remarks to the Author):

This is a well written, highly relevant article, which highlights the importance of social disadvantages and One Health factors associated with AMR in East Africa. Their focus on co-occurrence of risk factors, rather than competing or individualised risks provides an interesting insight. Furthermore, their data illustrating that antibiotic use is only one of many drivers of AMR and that a wider social contextualisation is required, is a much-needed statement. The suggestions made are to strengthen the authors points and support for their claims and to frame this work better within the context of available data.

- *Line 72-82: It is nice to see that other pertinent studies from East Africa have been highlighted. However, the authors state these are small, interfering that this study is somewhat larger. Of note, the studies mentioned provided more microbiological sampling than this study, and therefore it could be argued that is study was smaller in*

comparison. As stated in line 304, the sampling size of this study was too small to determine ABR rates by site or bacterial species, which would have been very useful; markedly improving the impact of the results. I would, therefore, not recommend focussing on comparison of size, but on comparison between approaches and results, highlighting the consistencies and differences in what they found/represent.

We have reworded the introduction to focus on the distinctive analytical approach that we take, rather than compare sample size.

- *Line 72-82: The authors then provide further comparison, stating that they standardise their results across more sites and countries and therefore are not limited by geography. Again, I would say that all these studies, including this one, are limited in their scope and geographical spread. This study does not address these geographic gaps, but simply provides a different sampling frame. I think this paragraph & the discussion would be improved if it the authors would focus on the limitations across One Health datasets, and the absence of standardised approaches or large-scale analysis that precludes comparison and prioritisation. They then might consider which approaches or minimum datasets would be required to enable better risk evaluation or future analysis in the discussion.*

As above, we have reworded the introduction to focus on the analytical approach we take. We also refer back to these studies in the discussion to suggest what we additionally learn from taking such an analytical approach.

- *Line 130: Was analysis completed on whether re-classifying intermediate to sensitive (rather than resistant) undertaken, and if so, did this markedly changed the results?*

We now refer to this sensitivity analysis in the methods, results and supplementary materials (Figs S8 and S9). The results were not markedly different. The recategorization reduced the proportion of MDR among patients from 47.6% to 37.7% (see Table S5), and in the profile regression analysis, this reduces the number of high-risk MDR clusters we detected from 10 to 6. However, largely the same variables emerge as important for determining the high-risk clusters. The profile of the high-risk clusters in terms of these variables is to a large extent unchanged. Therefore, we would argue that our results are robust, however intermediate ASTs are treated.

- *Lines 144-163 & 270: Here, I am unable to comment on whether the Bayesian approaches taken can show that the results “operate synergistically to shape ABR burden” (as they state in line 270), or whether in fact, other interpretations for these results could be inferred. I would appreciate it if further statistical opinion was sought as to whether this claim is entirely accurate.*

Given the analysis approach, the observed variables do operate jointly and therefore synergistically. Bayesian Profile Regression is exactly designed to explore these intersectional patterns, and this makes the analysis distinct from more traditional regression approaches.

- *Line 170: Kenya seems to have less ABR than the other countries. Would the authors like to state later on, why they think this is the case, and whether these data / differences correlate to other available data from these settings?*

Microbiological data allowing robust cross-country comparisons in this region (i.e. similar sampling and recruitment processes) are extremely sparse, our recently published paper ⁴

was intended to fill this gap. Therefore there are no reliable, comparable sources of data to compare these estimates to (outside of this study). Further, even if such standardised cross-country data were available, it's unlikely to have sampled from the same sites and clinics. However, we would speculate that there are two plausible possible reasons why the Kenyan distributions (proportion UTI, proportion MDR) look different to the other countries. Kenya is a more socioeconomically developed country, and the study recruited from the capital and surrounding areas (unlike Tanzania and Uganda), hence the patient sample is more urban, better educated and more affluent. In addition, due to pandemic restrictions Kenya sampled from proportionally fewer primary care facilities. This could also have affected the distributions observed, but it is difficult to draw strong conclusions.

- *Line 182 and Figure 4: Did HIV individuals have co-trimoxazole prophylaxis, which could have impacted on their results, and was this accounted for in 56% taking antibiotics? I also note that MDR UTI was higher in HIV (line 201). Again, did the role of CPT have any effect?*

We did not ask this specifically of HIV-positive individuals. The 56% captures those taking antibiotics for any reason, including UTI symptoms, so it was unfortunately not possible to unpick the role of co-trimoxazole prophylaxis in contributing to higher MDR rates in HIV-positive patients.

- *Line 236-239: Owning your own house was associated with a high-risk cluster, which might be argued serves as a proxy for higher SES, and contrary to the other information. Why do the authors think this result occurs and if this is contrary to other information, does it highlight a limitation of grouping results together?*

We agree this is a somewhat unusual result and highlights the need for a nuanced interpretation of both variables assumed to proxy for socio-economic status in this context and their correlations with other factors. One reason it might occur is that in clustering patients some of the variables which confer higher risk, e.g. owning livestock, are associated with agricultural livelihoods and rural living, which are more likely to also include owning land and housing as forms of wealth⁵. We have included a line about this in the discussion:

- *Line 242-244: Again, recognising AB packaging was associated with a high-risk cluster, whereas recognising the word antibiotic was associated with lower risk cluster. How do the authors justify these somewhat competing results?*

We observe these results also for predicted AB misuse⁶. Our interpretation rests on qualitative data analysed in that paper which demonstrated that familiarity with packaging is different to understanding the word medicalised term 'antibiotic', and they might have opposite socioeconomic status (and MDR) implications. Greater familiarity with packaging is more likely to occur with frequent exposure to sickness and antibiotics through clinics or drug shops. This is less likely to occur in better educated groups who are healthier for a range of reasons; whereas knowing the technical term 'antibiotic' (as opposed to local names like 'double colour') might relate to better education.

- *Lines 251: Interesting that EA ABR were similar, as was the analysis specifically on gram negative bacteria. Are we to assume then that these broad results are likely to represent those for gram-negative UTIs? If so, this should be stated.*

As the majority (n=1,009, 63%) of the observations are Gram negative, the overall results more broadly represent the situation in those type of pathogens. We added a short statement to this effect in the results.

- *Line 260: “Unprecedented”? As per my previous comments, this is merely a different analysis and sampling frame. Please remove.*

We edited this to ‘wide range’.

- *Line 268-300: The importance of WASH and environmental factors were clearly stated in the introduction, yet this message seems a little lost in the discussion. For example, animals, manure and waste dumping have higher ABR risks, and low risk clusters were associated with handwashing with soap. Whilst the authors mention “toilet features”, “hygiene”, “unprotected water” and “environmental waste” within the discussion, this could be written more cohesively to make it clearer to the reader that these variables are environmental/WASH health proxies, and that poverty and paucity of WASH are part of an interlinked / intersectional vulnerability.*

We have edited that section to describe how WASH vulnerabilities intersect with demographic, health and poverty dimensions.

- *Line 275-277: Nice to see it highlighted that ABR UTIs could provide a cycle of costs for the poorest / most vulnerable people. Consider adding whether this might, in fact, not just increase their costs, but also increase their ongoing risks through further healthcare exposures and AB exposures. Breaking these cycles is likely to have many impacts for these high-risk individuals.*

Thanks for the suggestion, we have added text to this effect.

- *Line 284-291: Overall a good section, but the authors might like to consider how specific pathways or delivery of healthcare systems could place vulnerable individuals at increased risks too, and consider adding this in.*

We have edited this section to hopefully highlight the role of under resourced healthcare systems to compound risks for vulnerable patients.

- *Line 291: ? “care burden patients”. Should this read care burden from ABR infections?*

This has been corrected.

Line 292-300: Excellent.

- *Line 305 (point 1): It is a shame that this study couldn’t analyse results by site and in particular, MDR species, as this would have allowed a more refined understanding of the key risks. This highlights once again the differences between this, compared with other studies, and an overarching issue with One Health data. That being, the balance between the level of sampling/ recruitment to enable granular yet affordable results. This enhanced data is often unfeasible in low-resource settings, due to costs, technical constraints or geographic inequities. Furthermore, One Health data is frequently skewed towards human health, prohibiting the integration with other datasets. I would very much appreciate if the authors could state more clearly the nuances of this limitation and consider the context of East Africa / One health. They might also want to briefly postulate any feasible approaches that could be considered to address this?*

Thanks for this; we have incorporated some words on this in the limitations – and reflected on the challenge and how our study could be used to guide the development of a affordable minimal surveillance dataset which takes account of these multifaceted drivers.

- *Line 305 (point 2): Consider adding in whether you can or cannot infer any directionality, especially around how poverty effects ABR risks.*

We cannot infer directionality very well with our data. We make a statement in the final conclusion about the need for longitudinal in-depth data to infer directionality and unpick processes of transmission from ABR development.

- *Line 305 (point 3): Another limitation is that some information obtained is from reported data, which is subject to bias, particularly around WASH behaviours (i.e. handwashing with soap).*

We have added a statement on self-reported behavioural data as a limitation.

- *Line 305 (point 4): Some aspects (i.e. WASH) have sparser questions (Table s5), providing only high-level data. It would be good to see how and why variables were chosen. Was this done using previously validated datasets or at the researcher's discretion / consortium consensus? Also, was this refinement done as part of the pilot period between 2017-18 (in protocol).*

We are limited partly by the questions in the questionnaire and household observations made, and had to make some trade-offs between depth and breadth. The scope of the tools and data collection were guided by the pilot work in 2017-18, but subsequent refinements made through piloting during 2018-19. These tools were developed through a 9-month iterative collaborative process within the consortium drawing on both literature review, and also diverse disciplines and local understandings (we consulted extensively with social scientists, clinicians, AMR specialists in these settings). We also ran a full piloting in the field – at this stage some more detailed questions had to be dropped to minimise length.

- *Line 312: "is only facilitating". Consider change to "is only associated with" for clarity to the reader.*

This section has now been edited and we discuss the need for detailed longitudinal data to unpick directionality and causality.

- *Discussion (general): Is there a plan to use genomic information to enhance or refine these results? Additionally, is further modelling planned within HATUA? If so, these would be important things to state.*

Thanks, yes this was just a first step to explore the plethora of variables we collected. The group intends to leverage the linked genomic data to refine measurement of ABR and explore how this relates to variables identified as important here. This is mentioned in the last sentence of the discussion.

- *Data (general): I found the preceding protocol paper and supplementary data highlighting the statistical methods very useful, and allows for open reproducibility. However, is there a reason why the primary (anonymised) data is not accessible via a secure repository, to enable further analysis / meta-analysis to be undertaken? This would be of wider benefit to the AMR community and enable cross country comparison with other datasets.*

Some of the microbiological data are available in WHONET, the full linked dataset (including genomic data) needs to be prepared for easy reuse. We are in the process of this, as well as writing key outputs from the study, and will make it available in a timely fashion in an online repository and according to the guidance from the funder. Meanwhile, people can access the data on reasonable request by contacting the PI or corresponding author (this is stated in the data availability statement)

- *Line 449. Please use “One Health” or “one Health” throughout for consistency.*

This has been changed throughout to ‘One Health’.

- *Figure 4: HH members work at hospitals have higher risk. Would the authors like to comment more on the role of healthcare exposure?*

We have reported this when describing Fig 4, but as this variable did not provide a clear signal in the subsequent profile regression, and our limited word count, we did not go into a lot of detail.

- *Supplementary data. Still has tracked changes in (for example S3l).*

This has been removed.

Reviewer #3 (Remarks to the Author):

- *1) The use of the term “integrated” is vague, and authors should specify what is meant or implied by Line 55: However, the contribution of these factors alongside antibiotic use is rarely investigated in an integrated way.*

This has been replaced by ‘holistic’.

- *2) There are some results that appear to be presented in the methods section and should be considered for moving to the results section.
Line 98 Less than 1% declined to participate, and this did not vary by site.
Line 109 We compared the characteristics of patients with positive UTI cultures (n=2,332), those with antibiotic susceptibility testing (AST) (n=2,063) and those who were included in the analysis sample (n=1,610) (see supplementary Table S2), and these varied little.*

We prefer to leave these very general features of the sampling in the methods section as they relate to the recruitment process. We prefer to keep the results for analysis which addresses the research questions.

- *3) Its unclear if the following is a previously published data point or a result of the study (in the methods section). If it is a result of the current analysis, consider moving to the results section.
Line 119: Gram-positive infections were more common in Kenya (48%) vs. Tanzania or Uganda (36% and 22% respectively). Susceptibility to the tested Abs...*

These results are drawn from earlier analysis and a reference has now been added (so we left this in the methods).

- *4) The importance of the assessment of antibiotic use in combination with all of the other variables in question was emphasized in the introduction. In the results, however, there was little mention of antibiotic use results. Consider expanding the*

amount of information given for this point in the results section, in addition to the text below.

Line 179: Treatment seeking behaviours for UTI (i.e., number of steps in seeking treatment, self-treatment, AB use, clinic attendance) was generally more complex in Tanzania and Ugandan patients than those in Kenya.

We have a very limited word count and need to summarise associations with many variables, hence preferred to concentrate on aspects with very clear strong associations with high-risk MDR clusters. The pathway variables have already been described and explored in detail here ⁷, and we have added this reference for readers to follow.

- *5) Consider improving the clarity of the following sentence by specifying what obstacles were observed (or measured) and what the results of those observations were.*

Line 199: MDR rates were higher if the household did not face obstacles accessing medicine, where a household member works in a hospital, and where they reported sharing antibiotics.

We have edited this to read “MDR rates were higher if the household reported that they did not face obstacles accessing medicine” to indicate this was self-report.

- *6) The following sentence is confusing and should be clarified*
Line 279: In our sample, behaviours and burden do not correspond, speaking to further injustice.

We have edited this part so that it reads: “In our sample, AB use behaviours and ABR burden do not correlate, speaking to further injustice. While lower education groups suffer the highest ABR burdens, despite being the least likely to misuse antibiotics.”

- *7) There is a double period at the end of the sentence in line 286 that should be edited.*

This has been edited.

Reviewer #4 (Remarks to the Author):

- *Despite the advantages of the approach used here, the results are still a bit challenging to distill, even with the helpful “heat map” provided. A wonderful addition to this paper would be to provide a joint (not just individual) comparison of predictive profiles regarding their effect on MDR UTI. For example, one could examine “typical” privileged, middle-class, and underprivileged covariate profiles, and compare their risks using the predictive profile features of the PReMiuM R-based package. These typical profiles must be defined a priori, from the literature or from the current data. Note that stitching together marginal averages of covariates may be inconsistent with the joint nature of the profile regression approach.*

Thank you for this useful suggestion.

We decided to calculate predictive profiles defined in two ways. First, we defined this a priori – driven by the literature on poverty and ABR risk⁸. We used the HATUA multidimensional poverty index developed in a previous publication⁶, which captures various domains of poverty using a standardised and validated technique commonly used in LMICs, and calculated profiles for patients falling within extremes of multidimensional poverty – the ‘very deprived’ and ‘not deprived’.

Second, we also defined predictive profiles for 'high risk' and 'low risk' patients from the current data by using the variables identified with clear signals in Figure 6 (these estimates are added to Fig 6). We compare the predictions and provide detailed posterior distributions broken down by site in the supplementary material (Figs S4 -S7).

There is a disparity in the predicted MDR UTI risk between the 'very deprived' and 'not deprived' considered profiles, and also between the 'high risk' and 'low risk' profiles as shown in Figure 6, but this is much larger when you use the extended list of variables identified through profile regression (31% vs 64%) compared with using very deprived/not deprived using standardised multidimensional poverty measures (57% vs 62%).

- *The outcome is MDR UTI – multi-drug resistant urinary tract infection – 3 or more categories of antibiotics. Did the authors utilize a sensitivity analysis using, say, 4 or more categories?*

For simplicity, we decided to stick with the recognised definition of 3 or more categories of AB, as doing so would simply mean reclassifying some people from being in the 1 outcome to 0, and we would expect the same pattern of associations. We have conducted additional sensitivity analyses by reclassifying the outcome according to AST intermediate outcome, and we observe a smaller number of high risk clusters, but the same variable associations.

- *“AB’s” (line 49) – should this be defined?*

This has now been defined.

- *“leap forward” – line 78 – ambiguous – reword*

This has been reworded as 'are more sophisticated in terms of'.

- *Line 85: “The data were ‘generated’”. This suggest that the data were synthetic. Is this what the authors mean?*

We have reworded as 'collected'.

- *Line 149: Which type of variable selection (binary or continuous)?*

The full technical details are mentioned in the supplementary material, section 4. We use variable selection formulation that involves cluster specific indicators, as proposed by Papathomas et al.⁹. (Full details are provided in the next comment).

- *How was a variable deemed to be important?*

The variable selection approach that involves cluster specific indicators, as proposed by Papathomas et al.⁹, results in estimating a continuous selection parameter for each variable, that is interpreted as the probability that the variable is important for the formation of the high and low risk clusters. The closer this probability is to 1, the more important the variable.

Related to the point above, we refer to the supplementary material, section 4:

“Due to the relatively large number of covariates included in the clustering model, it is of interest to determine which covariates are important for forming the clusters. A variable selection procedure is considered. We utilise the variable selection formulation that involves cluster specific indicators, as proposed by Papathomas et al.⁹This results in estimating continuous selection variables $\rho = (\rho_1, \dots, \rho_J)$. Note

that $\rho_j \in [0,1]$ for $j = \dots, 2, \dots, J$. The closer ρ_j is to 1 the corresponding covariate j is deemed to be important for determining the overall clustering structure. In contrast, the closer ρ_j is to 0, the corresponding covariate j is considered to be irrelevant for forming clusters. So, the ρ_j can be viewed as variable selection probabilities, where variable j is deemed important when the posterior mean or median of ρ_j exceeds some predefined threshold.”

- *Line 156: How much missing?*

We show the proportion of missing for each variable in Table S5, and this was highest for the variable ‘Using antibiotics when raising livestock’ (3.4%). We have edited the sentence to reassure the reader of the low level of missing data: “Missing responses for variables were treated as unknown random quantities, imputed during the Markov chain Monte Carlo inferential procedure in the same manner to model parameters (maximum proportion of missing was 3.4%)”.

- *Line 162: Rstudio is really just a front end to R, so please give the version of R and the version of PReMiuM used.*

This has been updated.

References

1. Sado, K. *et al.* Treatment seeking behaviours, antibiotic use and relationships to multi-drug resistance: A study of urinary tract infection patients in Kenya, Tanzania and Uganda. *PLOS Global Public Health* **4**, e0002709 (2024).
2. Asimwe, B. B. *et al.* Protocol for an interdisciplinary cross-sectional study investigating the social, biological and community-level drivers of antimicrobial resistance (AMR): Holistic Approach to Unravel Antibacterial Resistance in East Africa (HATUA). *BMJ Open* **11**, e041418 (2021).
3. Kahlmeter, G. EUCAST proposes to change the definition and usefulness of the susceptibility category 'Intermediate'. *Clinical Microbiology and Infection* **23**, 894–895 (2017).
4. Maldonado-Barragán, A. *et al.* Predominance of multidrug-resistant (MDR) bacteria causing urinary tract infections (UTIs) among symptomatic patients in East Africa: a call for action. 2023.06.13.23291274 Preprint at <https://doi.org/10.1101/2023.06.13.23291274> (2023).
5. Hadley, C., Maxfield, A. & Hruschka, D. Different forms of household wealth are associated with opposing risks for HIV infection in East Africa. *World Development* **113**, 344–351 (2019).
6. Green, D. L. *et al.* The role of multidimensional poverty in antibiotic misuse: a mixed-methods study of self-medication and non-adherence in Kenya, Tanzania, and Uganda. *The Lancet Global Health* **11**, e59–e68 (2023).
7. Keenan, K. *et al.* Unravelling patient pathways in the context of antibacterial resistance in East Africa. *BMC Infectious Diseases* **23**, 414 (2023).
8. Alividza, V. *et al.* Investigating the impact of poverty on colonization and infection with drug-resistant organisms in humans: A systematic review. *Infectious Diseases of Poverty* **7**, 76 (2018).
9. Papathomas, M., Molitor, J., Hoggart, C., Hastie, D. & Richardson, S. Exploring Data From Genetic Association Studies Using Bayesian Variable Selection and the Dirichlet

Process: Application to Searching for Gene \times Gene Patterns. *Genetic Epidemiology* **36**,
663–674 (2012).